# Dealing with Frequency Collapse in Time Series Embeddings by Post-Embedding reMapping

## Abstract

Transformer-based methods have made significant strides in time series forecasting tasks in recent years. However, we observe underfitting in numerous samples, e.g., pattern shifts or excessive deviation in extreme value regions, when testing the transform-based model that converges on the training set. Through the proposed spectral analysis of adjacent embedding sequences, we identify a frequency collapse issue in the embedding features generated by the top layer of the transformer backbone. To address this, we propose the Post-Embedding ReMapping (PErM) strategy that improves the frequency-domain representation of embeddings using fixed non-linear functions. Both two kinds of PErM functions that we insert into the model can effectively resolve the frequency collapse issue and lead to significant improvements in prediction performance. Experimental results show that our method outperforms state-of-the-art algorithms across multiple datasets. We will release our code after the review phase.

## 1 Introduction

Time series forecasting is a critical task across various real-world fields, including but not limited to finance, healthcare, weather, and supply chain management. Accurate time series forecasting allows organizations to make informed decisions, optimize processes, and anticipate future trends, making it a valuable tool for strategic planning and resource allocation. In recent years, with the rapid advancement of deep learning, an increasing number of learning-based time series forecasting methods have emerged (Zeng et al., 2023; Wang et al., 2024a; Xu et al., 2024b). In particular, following the remarkable success of transformer (Vaswani et al., 2017) in the field of natural language processing (Brown et al., 2020), a series of efforts have been made to adapt transformers more effectively for time series forecasting tasks (Wu et al., 2021; Nie et al., 2023; Liu et al., 2024b).

However, when testing a converged transformer model adapted for time series tasks, we still observe significant deviations between the predicted values and the patterns of the original time series, even in segments with clear periodic characteristics, and this deviation is particularly noticeable in the fitting of extreme value regions as shown in Fig. 1. Given the rapid signal variations in these regions, we hypothesize that these underfitting phenomena are related to the loss of certain high-frequency signals in the embeddings generated by the model.

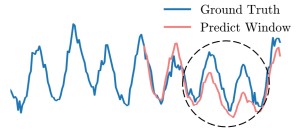

Figure 1: Example of underfitting.

Unfortunately, it is unreasonable to directly apply time-frequency analysis methods, such as spectral analysis, to embeddings of one time series sample. The model's forward process disrupts the internal temporal structure of the samples, thus the temporal information within each embedding is disordered. This means that performing spectral analysis directly on the embeddings is meaningless. In time series tasks, we observe that *the sliding window sampling method allows the sample sequence to directly reflect the internal temporal structure of each sample*. Thanks to this finding, we propose a spectral analysis method based on adjacent embedding sequences, and we further prove that the absence of high-frequency signals in the embedding sequence directly leads to the loss of high-frequency signals in the prediction target, which will result in underfitting.

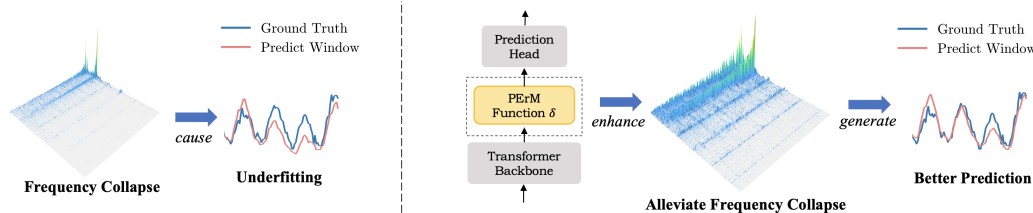

Figure 2: PErM helps to alleviate underfitting by enhancing frequencies in Embedding Sequence, thus improving the model's prediction performance.

Through our analysis, we confirm that training solely based on transformers can lead to a lack of high-frequency signals in the embeddings, which we refer to as *frequency collapse*. To address this issue, we propose the Post-Embedding reMapping (PErM) strategy. We introduce a pre-defined non-linear remapping layer between the top layer of the transformer backbone and the model's prediction head as shown in Fig. 2. This layer employs a non-linear mapping function to project high-frequency features from the original signals into a new frequency domain, while also re-integrating information within the embeddings. As the high-frequency features can be mapped to some lower freqencies in the new domain, this helps the model to capture high-frequency features more easily. Experiments show that using various PErM functions effectively alleviates the frequency collapse phenomenon and improves the final prediction performance.

In summary, we make the following contributions:

- We propose a framework for embedding-level spectral analysis using adjacent embedding sequences, and validate that the frequency collapse in the embedding sequence can lead to underfitting.

- We propose the Post-Embedding reMapping strategy to alleviate the issue of frequency collapse that arises during the training of the transformer on time series forecasting tasks, thereby improving the model's prediction accuracy.

- Compared with the state-of-the-art time series forecasters, our PErMformer achieves competitive performance across multiple time series forecasting datasets.

## 2 PRELIMINARIES AND RELATED WORKS

### 2.1 TIME SERIES FORECASTING TASKS

Given historical observations $x_1, x_2, \cdots, x_t$ with $t$ time steps, the forecasting task is to predict the values for the next $h$ time steps. In practice, we use a sliding window of length $n$ to generate historical observation samples for forecasters. These samples are also called *lookback windows*. Consider a time series: $\mathcal{X} = [x_1, x_2, x_3, \cdots, x_i, \cdots, x_T]^\top$, for each lookback window $\mathcal{X}_i = [x_{i-n+1}, x_{i-n+2}, \cdots, x_i]^\top$ sampled from $\mathcal{X}$, the prediction target is the data values for the upcoming $h$ time steps $\mathcal{Y}_i = [x_{i+1}, x_{i+2}, \cdots, x_{i+h}]^\top$.

### 2.2 RELATED WORKS

**Transformer-based Forecasters.** Numerous transformer-based time series forecasting models have emerged in recent years. PatchTST (Nie et al., 2023) introduced a standard paradigm for applying transformer models to time series forecasting, while some other methods like Crossformer (Zhang & Yan, 2023) and iTransformer (Liu et al., 2024b) tried to modify the architectures of the vanilla model to improve performance. Some transformer-based time series foundation models that emphasize zero-shot capabilities have also been proposed during the past year (Das et al., 2023; Liu et al., 2024c; Goswami et al., 2024).

**Fourier Feature Mapping.** Rahimi & Recht (2007) mapped the input data to a low-dimensional feature space using the Fourier transform to accelerate the training of kernel machines. Tancik et al.

(2020) applied this method in the image domain to enhance image resolution and demonstrated through experiments that this mapping function transforms the Neural Tangent Kernel into a stationary kernel in terms of effectiveness.

## 3 SPECTRAL ANALYSIS FROM THE PERSPECTIVE OF ADJACENT EMBEDDING SEQUENCE

Previous analyses of the representational capabilities of time series models have often focused on the *per-sample* level, which involves examining and visualizing the structure of the embedding feature generated from a single sample. However, consider the mapping function $f_\theta : \mathcal{X} \to \mathcal{E}$ given by the neural networks parameterized by $\theta$, the mapped embedding $\mathcal{E}$ do not necessarily preserve the order of features within the sample $\mathcal{X}$. Due to the disruption of the positional relationships of the raw time series values, the embedding can hardly retain the understandable temporal structures. Thus it's challenging to assess whether the mapping function $f_\theta$ can effectively represent the characteristics of the original time series, such as periodicity and trends.

Fortunately, we notice a notable characteristic of time series tasks is the strong correlation between adjacent samples, which is different from other tasks. We concatenate consecutive samples to generate the adjacent sample sequence $\mathcal{X}_{seq}$:

$$\mathcal{X}_{seq_{i,j}} = [\mathcal{X}_i, \mathcal{X}_{i+1}, \cdots, \mathcal{X}_{i+j}] = \begin{bmatrix} x_i & x_{i+1} & \cdots & x_{i+j} \\ x_{i+1} & x_{i+2} & \cdots & x_{i+j+1} \\ \vdots & \vdots & \ddots & \vdots \\ x_{i+n} & x_{i+1+n} & \cdots & x_{i+j+n} \end{bmatrix} \tag{1}$$

It can be observed that **this sample sequence inherits the characteristics of the original time series along the temporal dimension**, as each row of the sequence (highlighted in red) is one subset of the raw time series (along the direction highlighted in blue). According to this observation, we consider analyzing the adjacent embedding sequence $\mathcal{E}_{seq_{i,j}} = [\mathcal{E}_i, \mathcal{E}_{i+1}, \cdots, \mathcal{E}_{i+j}]$ generated from the adjacent sample sequence $\mathcal{X}_{seq_{i,j}}$ by $f_\theta$. Given that the temporal structure of the sample sequence is equivalent to that of the original time series segment, and that the mapping from the sample sequence to the embedding sequence preserves the relative positional relationships over time, we can investigate the temporal variations of each feature within the embedding sequence. Specifically, for the $k$-th feature, we can perform spectral analysis on $\mathcal{E}_{seq_{i,j}}^k$ to obtain its energy distribution across different frequencies, thereby determining whether the mapping $f_\theta$ on this dimension can capture periodic correlations, and if so, whether it captures high-frequency or low-frequency components. This analysis framework, as shown in Fig. 3, helps to provide insights into the model's capacity to effectively represent the time-frequency information inherent in the original sequence in an interpretable manner.

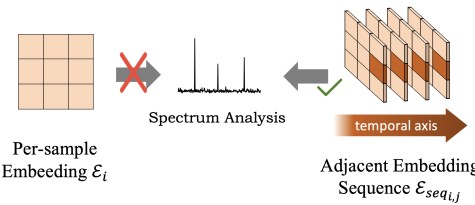

Per-sample
Embeeding $\mathcal{E}_i$

Spectrum Analysis

temporal axis

Adjacent Embedding
Sequence $\mathcal{E}_{seq_{i,j}}$

Figure 3: Per-sample Embedding Analysis v.s. Adjacent Embedding Sequence Analysis. The temporal structure in per-sample embedding is disrupted, while the adjacent embedding sequence reserves the temporal structure.

From the perspective of adjacent embedding sequence, the $k$-th feature in $\mathcal{E}$ can be represented by the stochastic process $\mathcal{E}^k(t)$, and $\mathcal{E}_i$ is the collection of the samples from all stochastic processes. We conduct a preliminary exploration of the spectral relationship between $\mathcal{E}_i$ and the corresponding prediction target $\mathcal{Y}_i$.

To simplify the problem, suppose that all $\mathcal{E}(t)$ are stationary stochastic processes and uncorrelated with each other, and $\mathcal{E}_i$ is mapped to $\mathcal{Y}_i$ via a known deterministic matrix $W$ (the prediction head). We prove that if the stochastic processes where $\mathcal{E}_i$ is sampled lack high-frequency information, the theoretical upper limit of the high-frequency information of $\mathcal{Y}_i$ will decrease.

Given the stochastic processes $\mathcal{Y}^k(t)$ mapped from $\mathcal{E}^k(t)$ by $W$, consider the power spectral density $S_{\mathcal{Y}^k}(\omega)$ of $\mathcal{Y}^k(t)$ at frequency $\omega$, we have:

$$S_{\mathcal{Y}^k}(\omega) = \lim_{T \to +\infty} \left\{ \frac{1}{2T} \mathbb{E} \left| \sum_{l=1}^{n} w_{k,l} \int_{-T}^{T} \mathcal{E}^l(t) \exp(-j\omega t) dt \right|^2 \right\} \tag{2}$$

Since each stochastic processes in $\mathcal{E}(t)$ are uncorrelated with each other, we have:

$$S_{\mathcal{Y}^k}(\omega) = \sum_{l=1}^{n} w_{k,l}^2 \left\{ \lim_{T \to +\infty} \frac{1}{2T} \int_{-T}^{T} \int_{-T}^{T} R_{\mathcal{E}^l}(t-s) \exp(-j\omega(t-s)) dt ds \right\} \tag{3}$$

where $R_{\mathcal{E}^l}(\cdot)$ denotes the autocorrelation function of the stationary stochastic process $\mathcal{E}^l(t)$. Let $\tau = t - s$, and we further get:

$$S_{\mathcal{Y}^k}(\omega) = \sum_{l=1}^{n} w_{k,l}^2 \int_{-\infty}^{+\infty} R_{\mathcal{E}^l}(\tau) \exp(-j\omega\tau) d\tau = \sum_{l=1}^{n} w_{k,l}^2 S_{\mathcal{E}^l}(\omega) \tag{4}$$

where $S_{\mathcal{E}^l}(\omega)$ represents the power spectral density of the stochastic process signal $\mathcal{E}^l(t)$ at frequency $\omega$. The detailed derivation process is listed in Appendix A. This proves that the spectral density of $\mathcal{Y}^k$ at frequency $\omega$ is a linear combination of the spectral densities of each $\mathcal{E}^l$ at frequency $\omega$. Since the sample sequence exhibits the same temporal structure in both $x$ and $y$ dimensions, we can consider $\mathcal{Y}_i$ as a sequence sampled from $\mathcal{Y}^k(t)$. Thus we can conclude that if the stochastic process corresponding to the embedding sequence $\mathcal{E}_{seq_{i,j}}$ does not contain high-frequency information, then the $\mathcal{Y}_i$ obtained by linearly transforming the embeddings will also not contain high-frequency information.

## 4 UNDERFITTING AND FREQUENCY COLLAPSE

We observe that when training a time series model directly on the Transformer backbone, the model appears to exhibit underfitting when handling samples with rich detailed features. As shown in Fig. 4, the trend of the curve changed greatly from timestamp 80 to timestamp 120, where the model performs poorly. To further investigate, we extract the embeddings given by the top transformer block before the model's prediction head and conduct spectral analysis on the adjacent embedding sequence around this sample. Via Discrete Fourier Transform (DFT), each zero-centered feature dimension in the embedding sequence is mapped to the frequency domain. We then sort the features in descending order based on their maximum amplitude in the frequency domain, and

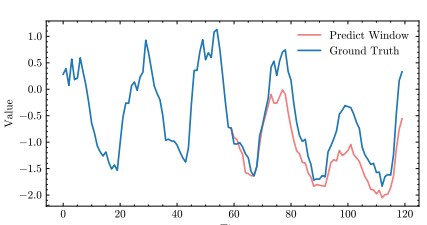

Figure 4: Example of underfitting when employing a converged model.

finally obtain the feature spectrogram of this sequence ( 5a). We also visualize the spectrogram obtained from the union sequence of the corresponding prediction targets through DFT ( 5b).

Comparing two spectrograms, we can observe that the original sequence exhibits rich high-frequency details. Even at frequencies above 50, several frequencies also show significantly higher amplitudes than others, while in the spectrogram of the embedding sequence, there is an absence of notable amplitudes at the corresponding frequencies. Rahaman et al. (2019b) found the *spectral bias* phenomenon that neural networks favor the low-frequency information during the training process. However, in the context of time series forecasting, a converged model appears to excessively overlook the modeling of high-frequency features. As a result, the embedding sequence contains little high-frequency information which is crucial for accurately fitting autoregressive values. According to Sec. 3, this eventually leads to the loss of high-frequency components in the prediction series, thereby causing several underfitting samples. We refer to this phenomenon as **frequency collapse**.

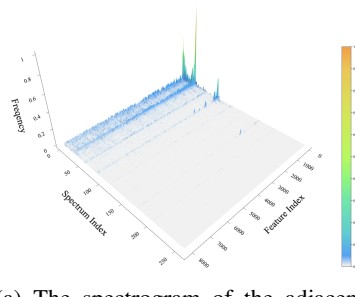
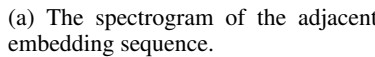
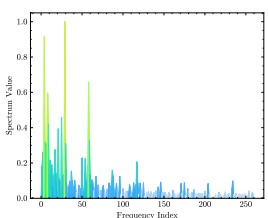

(a) The spectrogram of the adjacent embedding sequence.

(b) The spectrogram of the sample.

Figure 5: Two spectrograms corresponding to the example in Fig. 4.

**Empirical Study.** We experiment to empirically demonstrate how frequency collapse in the embedding sequence impacts the final prediction results. Within each test iteration, we apply a low-pass filter (20% cutoff frequency) to the embedding sequence generated by PErMformer and then generate the final predictions based on the filtered embeddings. As shown in Fig. 6, we visualize the prediction results before and after cutting off frequency. By comparison, we observe that when the embeddings lack high-frequency information, the resulting predictions exhibit significant underfitting. The predicted sequence shows a noticeable pattern shift from the original sequence in several areas (around time point 175), and the model's ability to fit extreme value is markedly reduced (around time points 100 and 125).

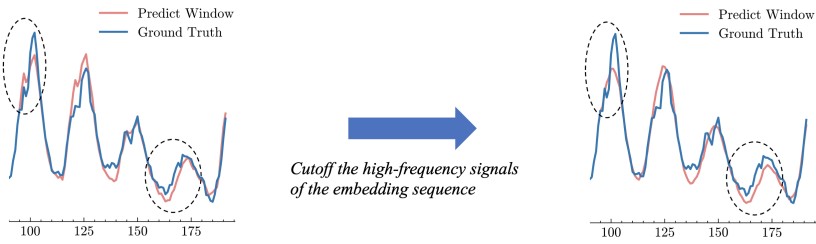

Figure 6: One sample on ECL dataset. The absence of high-frequency information in the embedding sequences leads to a loss of detailed features in the predicted values.

## 5 METHODS

In this section, we introduce the base model of our work and the post-embedding remapping strategy which helps to solve the frequency collapse problem. The architecture of our work is presented in Fig. 7

### 5.1 GENERAL TIME SERIES TRANSFORMER MODEL

In the processing of time series features, we adopt several well-known and empirically validated strategies specifically designed for time series Transformers. Meanwhile, the encoder in our model employs a pure Transformer backbone architecture. We will introduce these processing strategies and the backbone architecture in detail:

**Channel Independence (CI).** The Channel Independence strategy proposed by Nie et al. (2023) splits the multi-dimensional time series data into multiple one-dimensional sequences, which are then fed into the model for training separately. This approach decouples the model structure from the number of time series dimensions, enabling the model to simultaneously utilize time series data from datasets with varying dimensionalities for training, thereby providing a consistent data standardization for building time series foundation models.

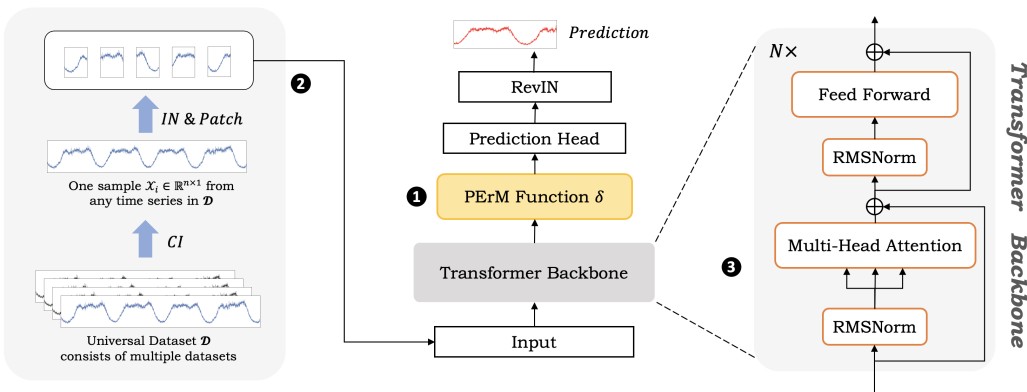

Figure 7: The architecture of PErMformer. ❶ The post-embedding reMapping (PErM) layer is introduced to alleviate the frequency collapse issue in the embedding sequence. ❷ The data processing follows the current mainstream foundation model's data processing framework to validate the effectiveness of the PErM strategy when facing diverse data distributions. ❸ The network structure of the Transformer backbone.

**Reversible Instance Normalization (IN & RevIN).** We employ the Reversible Instance Normalization (RevIN) strategy ( Kim et al. (2022)) in our base model. Specifically, for each one-dimensional time series sample, we normalize the sample before feeding it into the network and then denormalize the model output sequences (prediction targets). This helps to suppress non-stationary information in the intermediate layers, thereby mitigating the impact of data bias on the model's performance.

**Patching.** We segment each time series sample into multiple patches, with each patch corresponding to a token, and input these token sequences into the model. In contrast to treating each data point as a token, this approach not only effectively enhances the model's performance but also significantly reduces the inference time due to the reduction in the token count by a factor of the patch length.

**Transformer Backbone.** As the field of general pre-trained transformers continues to evolve, the network architecture of transformer blocks has also been undergoing continuous refinement. Several language models, including LLaMA (Touvron et al., 2023), Mistral (Jiang et al., 2023), and Gemma (Team et al., 2024), have introduced improvements over traditional structures. We believe that the rationale behind these architectural enhancements is applicable to general tasks. Consequently, we choose the latest Gemma transformer block as our backbone. This block employs the pre-Norm strategy and enhances the computational efficiency of attention through a matrix merging mechanism.

Since time series data inherently exhibits autocorrelation, employing positional embedding may disrupt this autocorrelation. Therefore, we abandon the Positional Embedding layer and remove the Rotary Position Embedding (RoPE) module from the selected transformer block.

## 5.2 POST-EMBEDDING REMAPPING

The key point to address the above frequency collapse issues is guiding the model to enrich the embeddings with high-frequency representations that are correlated with the original time series. To this end, we propose the Post-Embedding reMapping (PErM) strategy. Specifically, we introduce a non-trainable nonlinear mapping function $\delta(\mathcal{E})$ between the transformer encoder and the prediction head. This function can alleviate frequency collapse in embeddings by mapping the challenging learning objectives that contain high-frequency information into a combination of easier-to-learn low-frequency learning objectives during the training process.

We explore two kinds of mapping functions under the PErM strategy:

**Random Fourier Feature Mapping (RFF).** In this case (Fig. 8a), we use

$$\delta(\mathcal{E}) = [\cos(R\mathcal{E}), \sin(R\mathcal{E})]$$

as the mapping layer, where $R$ is a square matrix sampled from a Gaussian Distribution. As the feature dimension has become twice as large as before, we introduce a trainable matrix $W$ to project $\delta(\mathcal{E})$ back to the dimension of $\mathcal{E}$. The random matrix $R$ maps the embeddings into other feature spaces, introducing diversity and stochasticity into the representations. The sine and cosine functions make the mapping non-linear. Since Tancik et al. (2020) has demonstrated that using the random Fourier feature is an effective way to augment raw image features, we attempt to explore whether this mapping can guide the model to more effectively learn high-frequency representations of time series in the embedding feature space generated at the top layer.

**Layers of Pretrained Transformers (LPT).** Recent works have shown that leveraging the layers of pre-trained transformers, especially the layers of large language models, can effectively boost the performance of non-language tasks, such as image classification and logical reasoning tasks ( Lu et al. (2022); Dinh et al. (2022); Pang et al. (2024)). This might indicate that the parameter connections within large language models possess a certain degree of generalizable computational capability. Inspired by this, we view the layer of the large language model's network as highly abstract non-linear mapping functions, i.e.,

$$\delta(\mathcal{E}) = FPTL(\mathcal{E})$$

where $FPTL$ denotes Frozen Pretrained Transformer Layer (Fig. 8b). We align the dimension of the embeddings with the dimension of the pre-trained model's hidden space through a trainable Pre-FPTL adapter layer. After passing through the mapping function, we map the embeddings back to their original dimension using a trainable Post-FPTL adapter layer.

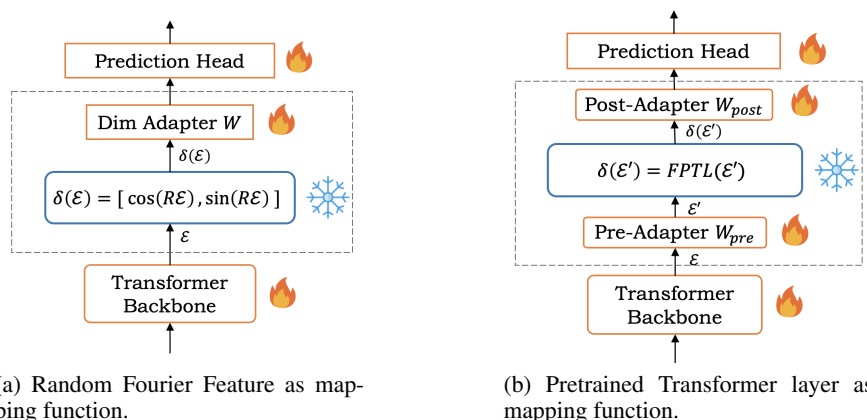

(a) Random Fourier Feature as mapping function.

(b) Pretrained Transformer layer as mapping function.

Figure 8: The network structures of the model when adopting the two PErM functions. Parameters in boxes labeled with flame are trainable during the training process, while those in boxes labeled with snowflake are frozen (non-trainable).

## 6   THE EFFECTIVENESS OF PERM STRATEGY.

In this section, we will validate the effectiveness of the PErM strategy through experiments, as well as evaluate the performance of different PErM strategies and settings.

### 6.1   SETUPS

**Datasets.** We adopt seven well-known and publicly available datasets to benchmark long-term forecasting models, including ECL, ETTh1, ETTh2, ETTm1, ETTm2, Traffic, and Weather. All datasets are divided into training, validation, and test sets with a 7:1:2 ratio. For further information about the datasets, please refer to Appendix E.

**Model lists.** We design three models based on the PErMformer architecture with 12 transformer blocks in Fig. 7. These models differ only in terms of the PErM strategy employed.

- *PErM-None.* The model does not employ any PErM strategy. The embedding given by the Transformer block is directly fed into the prediction head.

- *PErM-RFF.* The model employs random Fourier feature mapping as the PErM function.

- *PErM-LPT.* The model employs the layer of the pre-trained transformer as the PErM function. By default, we use the top layer of Gemma-2B as the mapping function.

**Task.** The prediction task is to predict the next 96 sequence values based on a given lookback window of length 512. Each model is trained on the Universal dataset $\mathcal{D}$ composed of the aforementioned seven datasets in an autoregression manner. Mean Square Error (MSE) is used as the evaluation metric.

## 6.2 RESULTS AND ANALYSIS

The prediction results of the three variants are listed in Table 1. Compared to models that do not use the PErM strategy, the two models employing different PErM strategies show significant improvements in prediction accuracy across almost all datasets. Specifically, using RFF as the PErM strategy results in an average reduction of 0.0208 in MSE, while using Gemma-2B top layer as the PErM strategy leads to an average reduction of 0.0217 in MSE. Additionally, we observe that for datasets with more pronounced periodic feature variations (ETTh1 & ETTh2), the use of the PErM strategy leads to more substantial improvements in performance.

Table 1: Forecasting results of the three models on seven datasets in MSE. The best result is highlighted in bold. Imp. denotes the performance improvement comparing with the method without the PErM strategy (PErM-None).

|  | ECL | ETTh1 | ETTh2 | ETTm1 | ETTm2 | Traffic | Weather |
|---|---|---|---|---|---|---|---|
| PErM-None | 0.136 | 0.360 | 0.340 | 0.300 | 0.198 | 0.377 | 0.171 |
| PErM-RFF | **0.130** | 0.320 | 0.305 | 0.279 | **0.188** | 0.361 | **0.153** |
| Imp. | 4.4% ↑ | 11.1% ↑ | 10.2% ↑ | 7.0% ↑ | 5.1% ↑ | 4.2% ↑ | 10.5% ↑ |
| PErM-LPT | 0.131 | **0.317** | **0.299** | **0.275** | 0.191 | **0.361** | 0.156 |
| Imp. | 3.7% ↑ | 11.9% ↑ | 12.1% ↑ | 8.3% ↑ | 3.5% ↑ | 4.2% ↑ | 8.8% ↑ |

Additionally, we analyze the spectrograms of the adjacent embedding sequences generated by different models in the dataset. Fig. 9 shows the prediction results of the three models on the same lookback window, along with the corresponding spectrograms of the embedding sequences for that period. We can observe that compared to the PErM-None method, the two PErM strategies we designed effectively mitigate the phenomenon of frequency collapse. The spectrograms of the adjacent embedding sequences for both strategies exhibit distinct and diverse peaks at high frequencies. According to the prediction results, the PErM-None method shows significant drift compared to the original sequence, while the PErM-RFF and PErM-LPT methods more accurately fit the characteristics of the original curve. More cases can be found in Appendix H.

We also need to emphasize that the presence of high-frequency information is more important than the specific distribution characteristics of the high-frequency information. There seems to be no direct correlation between the specific distribution of high-frequency features and the final performance. Although both the PErM-RFF and PErM-LPT methods perform well in prediction, their embedding sequences exhibit different characteristics in the high-frequency range. This is because the mapping relationship between the high-frequency features of the embedding sequence and the prediction target depends on the trainable parameters in the prediction head matrix $W$.

## 6.3 EXPLORATION OF SETTINGS FOR PERM

We explore the impact of various settings on the effectiveness of the PErM strategy, including whether to freeze the parameters in the mapping function or not, and whether post-embedding remapping or pre-embedding remapping proves to be more effective.

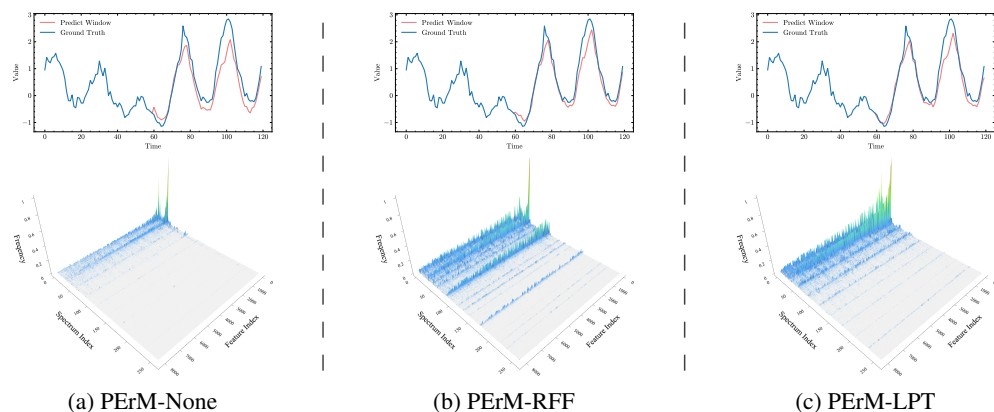

(a) PErM-None                    (b) PErM-RFF                    (c) PErM-LPT

Figure 9: The prediction results and the corresponding spectrograms of the embedding sequences of the three models on the same lookback window on dataset ECL.

**Trainable or Non-trainable?** We investigate the impact of freezing the parameters of the PErM layer or not on the prediction results through experiments. According to the experimental results (Table 2), on both PErM strategies, using non-trainable parameters can achieve better performance than tuning the parameters (PErM-XX-tune). We infer that the training process might cause the trainable mapping function to converge to simpler solutions. Additionally, we find that directly fine-tuning the pre-trained transformer layer may lead to non-convergence of the results during the experiments.

**Post-Embedding reMapping or Pre-Embedding reMapping?** We further investigate the relationship between the position of the remapping function in the model and the prediction performance. Different

Table 2: Ablation study on settings of PErM strategy.

| Model | Avg. MSE |
|---|---|
| PErM-RFF | 0.248 |
| PErM-RFF-tune | 0.253 |
| PreErM-RFF | 0.260 |
| PErM-LPT | 0.247 |
| PErM-LPT-tune | 0.266 |
| PreErM-LPT | 0.252 |

from post-embedding remapping which adds the mapping layer after the model's top layer, pre-embedding remapping adds the mapping layer before the model's bottom layer, directly transforming the raw time series features. Experiments show that the PostErM strategy is superior to the PreErM (PreErM-XX) strategy for time series prediction (Table 2). We speculate that the high-frequency information produced by PreErM may be lost during the forward process, resulting in less rich representations compared to those generated by PostErM.

As an extension of the PErM-FPT strategy, we also explore the impact of using different layers from various large language models as mapping functions. For more details, please refer to the Appendix G.

## 7 COMPARISON WITH THE SOTA METHODS

By default, our proposed method *PErMformer* denotes the *PErM-LPT* model mentioned above.

**Baselines.** We compare our method with several state-of-the-art methods that are designed for long-term time series forecasting tasks. These baselines include a series of Transformer-based models, like Informer (Zhou et al., 2021), Autoformer (Wu et al., 2021), FEDformer (Zhou et al., 2022), Pyraformer (Liu et al., 2021), Non-stationary Transformer (Liu et al., 2022), PatchTST (Nie et al., 2023), MICN (Wang et al., 2023), Crossformer (Zhang & Yan, 2023), UniTime Liu et al. (2024a), iTransformer (Liu et al., 2024b), and Time-LLM Jin et al. (2024). Some other competitive models are also selected as our baseline. These methods include TimesNet Wu et al. (2023), DLinear (Zeng et al., 2023), TSMixer (Ekambaram et al., 2023), FITS (Xu et al., 2024b), and TimeMixer (Wang et al., 2024a). More implementation details are listed in Appendix F.2.

Table 3: Performance comparison with nine competitive long-term time series forecasting methods. The best results are highlighted in red, and the second-best results are highlighted in blue.

| Methods | | PErMformer MSE | PErMformer MAE | FITS MSE | FITS MAE | iTransformer MSE | iTransformer MAE | PatchTST MSE | PatchTST MAE | UniTime MSE | UniTime MAE | TSMixer MSE | TSMixer MAE | TimeMixer MSE | TimeMixer MAE | DLinear MSE | DLinear MAE | MICN MSE | MICN MAE | TimesNet MSE | TimesNet MAE |
|---|---|---|---|---|---|---|---|---|---|---|---|---|---|---|---|---|---|---|---|---|---|
| ECL | 96 | 0.131 | 0.228 | 0.144 | 0.242 | 0.148 | 0.239 | 0.136 | 0.273 | 0.183 | 0.274 | 0.204 | 0.308 | 0.156 | 0.247 | 0.14 | 0.237 | 0.165 | 0.276 | 0.168 | 0.272 |
| | 192 | 0.149 | 0.244 | 0.164 | 0.267 | 0.167 | 0.258 | 0.153 | 0.28 | 0.189 | 0.28 | 0.218 | 0.329 | 0.17 | 0.261 | 0.154 | 0.25 | 0.176 | 0.288 | 0.187 | 0.289 |
| | 336 | 0.166 | 0.263 | 0.222 | 0.339 | 0.178 | 0.271 | 0.204 | 0.296 | 0.203 | 0.294 | 0.239 | 0.35 | 0.187 | 0.278 | 0.169 | 0.268 | 0.186 | 0.297 | 0.203 | 0.304 |
| | 720 | 0.218 | 0.312 | 0.3 | 0.403 | 0.209 | 0.298 | 0.246 | 0.328 | 0.242 | 0.325 | 0.272 | 0.373 | 0.228 | 0.312 | 0.204 | 0.3 | 0.208 | 0.317 | 0.254 | 0.343 |
| ETTh1 | 96 | 0.317 | 0.371 | 0.334 | 0.376 | 0.337 | 0.383 | 0.365 | 0.409 | 0.343 | 0.392 | 0.414 | 0.46 | 0.331 | 0.376 | 0.352 | 0.402 | 0.337 | 0.396 | 0.347 | 0.395 |
| | 192 | 0.356 | 0.394 | 0.379 | 0.408 | 0.383 | 0.412 | 0.409 | 0.441 | 0.385 | 0.419 | 0.516 | 0.531 | 0.374 | 0.404 | 0.391 | 0.431 | 0.401 | 0.447 | 0.401 | 0.43 |
| | 336 | 0.372 | 0.407 | 0.397 | 0.414 | 0.432 | 0.439 | 0.46 | 0.473 | 0.42 | 0.44 | 0.618 | 0.597 | 0.427 | 0.431 | 0.398 | 0.435 | 0.407 | 0.452 | 0.431 | 0.445 |
| | 720 | 0.373 | 0.425 | 0.388 | 0.433 | 0.443 | 0.465 | 0.626 | 0.559 | 0.415 | 0.454 | 0.706 | 0.665 | 0.415 | 0.438 | 0.461 | 0.498 | 0.524 | 0.535 | 0.438 | 0.464 |
| ETTh2 | 96 | 0.299 | 0.35 | 0.3 | 0.347 | 0.31 | 0.354 | 0.33 | 0.379 | 0.297 | 0.345 | 1.125 | 0.823 | 0.305 | 0.351 | 0.299 | 0.358 | 0.362 | 0.406 | 0.343 | 0.371 |
| | 192 | 0.381 | 0.396 | 0.374 | 0.394 | 0.39 | 0.402 | 0.387 | 0.421 | 0.377 | 0.395 | 2.831 | 1.461 | 0.382 | 0.397 | 0.394 | 0.421 | 0.516 | 0.492 | 0.404 | 0.41 |
| | 336 | 0.416 | 0.427 | 0.401 | 0.42 | 0.434 | 0.437 | 0.451 | 0.459 | 0.419 | 0.429 | 2.973 | 1.5 | 0.434 | 0.436 | 0.507 | 0.493 | 0.603 | 0.538 | 0.462 | 0.455 |
| | 720 | 0.403 | 0.435 | 0.416 | 0.437 | 0.433 | 0.446 | 0.548 | 0.521 | 0.429 | 0.446 | 2.379 | 1.293 | 0.488 | 0.476 | 0.911 | 0.681 | 0.895 | 0.675 | 0.471 | 0.473 |
| ETTm1 | 96 | 0.275 | 0.345 | 0.266 | 0.325 | 0.29 | 0.349 | 0.276 | 0.332 | 0.326 | 0.369 | 0.369 | 0.423 | 0.278 | 0.339 | 0.261 | 0.329 | 0.27 | 0.346 | 0.33 | 0.37 |
| | 192 | 0.323 | 0.375 | 0.3 | 0.347 | 0.325 | 0.368 | 0.316 | 0.364 | 0.366 | 0.39 | 0.393 | 0.446 | 0.318 | 0.361 | 0.293 | 0.35 | 0.306 | 0.373 | 0.372 | 0.393 |
| | 336 | 0.336 | 0.386 | 0.333 | 0.368 | 0.36 | 0.392 | 0.344 | 0.381 | 0.396 | 0.411 | 0.455 | 0.491 | 0.337 | 0.381 | 0.323 | 0.368 | 0.352 | 0.408 | 0.372 | 0.404 |
| | 720 | 0.432 | 0.44 | 0.422 | 0.432 | 0.425 | 0.431 | 0.403 | 0.422 | 0.452 | 0.443 | 0.595 | 0.583 | 0.399 | 0.42 | 0.378 | 0.406 | 0.392 | 0.433 | 0.46 | 0.453 |
| ETTm2 | 96 | 0.191 | 0.27 | 0.181 | 0.262 | 0.191 | 0.272 | 0.188 | 0.272 | 0.179 | 0.266 | 0.257 | 0.361 | 0.183 | 0.261 | 0.173 | 0.262 | 0.192 | 0.285 | 0.19 | 0.266 |
| | 192 | 0.25 | 0.313 | 0.244 | 0.301 | 0.264 | 0.318 | 0.254 | 0.317 | 0.246 | 0.309 | 0.565 | 0.59 | 0.251 | 0.305 | 0.306 | 0.314 | 0.306 | 0.369 | 0.266 | 0.313 |
| | 336 | 0.289 | 0.338 | 0.303 | 0.343 | 0.328 | 0.358 | 0.295 | 0.341 | 0.312 | 0.35 | 0.963 | 0.751 | 0.311 | 0.348 | 0.3 | 0.355 | 0.37 | 0.408 | 0.332 | 0.351 |
| | 720 | 0.389 | 0.396 | 0.421 | 0.413 | 0.428 | 0.411 | 0.41 | 0.416 | 0.413 | 0.409 | 2.463 | 1.302 | 0.409 | 0.402 | 0.512 | 0.483 | 0.521 | 0.493 | 0.435 | 0.411 |
| Traffic | 96 | 0.361 | 0.256 | 0.411 | 0.281 | 0.392 | 0.268 | 0.368 | 0.257 | 0.477 | 0.309 | 0.531 | 0.358 | 0.476 | 0.297 | 0.413 | 0.287 | 0.522 | 0.313 | 0.589 | 0.315 |
| | 192 | 0.377 | 0.262 | 0.423 | 0.286 | 0.413 | 0.277 | 0.385 | 0.265 | 0.479 | 0.31 | 0.563 | 0.385 | 0.508 | 0.301 | 0.424 | 0.29 | 0.537 | 0.318 | 0.617 | 0.326 |
| | 336 | 0.392 | 0.272 | 0.434 | 0.291 | 0.425 | 0.283 | 0.396 | 0.272 | 0.494 | 0.316 | 0.58 | 0.393 | 0.516 | 0.308 | 0.438 | 0.299 | 0.55 | 0.323 | 0.635 | 0.338 |
| | 720 | 0.453 | 0.316 | 0.463 | 0.308 | 0.458 | 0.3 | 0.439 | 0.3 | 0.528 | 0.333 | 0.619 | 0.418 | 0.547 | 0.323 | 0.466 | 0.316 | 0.576 | 0.334 | 0.659 | 0.349 |
| Weather | 96 | 0.156 | 0.21 | 0.146 | 0.198 | 0.178 | 0.219 | 0.158 | 0.214 | 0.172 | 0.215 | 0.18 | 0.252 | 0.162 | 0.208 | 0.174 | 0.233 | 0.193 | 0.25 | 0.169 | 0.219 |
| | 192 | 0.206 | 0.257 | 0.19 | 0.24 | 0.226 | 0.259 | 0.213 | 0.266 | 0.219 | 0.255 | 0.218 | 0.287 | 0.208 | 0.251 | 0.218 | 0.278 | 0.24 | 0.301 | 0.225 | 0.265 |
| | 336 | 0.262 | 0.296 | 0.242 | 0.279 | 0.282 | 0.3 | 0.248 | 0.284 | 0.274 | 0.295 | 0.261 | 0.321 | 0.264 | 0.293 | 0.263 | 0.314 | 0.282 | 0.301 | 0.282 | 0.304 |
| | 720 | 0.327 | 0.344 | 0.322 | 0.338 | 0.358 | 0.35 | 0.32 | 0.338 | 0.352 | 0.345 | 0.318 | 0.363 | 0.345 | 0.346 | 0.332 | 0.374 | 0.35 | 0.387 | 0.359 | 0.354 |
| $1^{st}$ | | 13 | 13 | 6 | 10 | 0 | 2 | 1 | 0 | 1 | 1 | 1 | 0 | 0 | 1 | 6 | 1 | 0 | 0 | 0 | 0 |

**Evaluation Metrics and Tasks Settings.** Following the common practice in most recent time series forecasting works, we employ Mean Square Error (MSE) and Mean Absolute Error (MAE) as the evaluation metrics. For each dataset, we compare our method with baselines using four prediction horizons $H \in \{96, 192, 336, 720\}$. The length of the lookback window $L$ is set to 512 for PErMformer, while that for the baseline is set according to their settings in the paper. Following the practice of UniTime, we combine all datasets together to train a unified model and test separately on each dataset. The dataset setup is widely used for foundation models, and it imposes higher demands on the model's fitting and generalization capabilities. This means *our method uses only one set of hyperparameters for all datasets*, while most of the baselines specifically choose a set of hyperparameters for each dataset.

**Platform.** All experiments are conducted on the server equipped with 12 Intel(R) Xeon(R) Gold 5317 CPUs @ 3.00GHz and 4 NVIDIA GA102 GeForce RTX 3090 GPUs. The operating system is Ubuntu 22.04 LTS. Each experiment is conducted with one GPU.

**Results.** Due to the space limitations, we select a few more competitive methods as baselines in Table 3 and use MSE and MAE as the evaluation metric. The full results of all baselines are provided in the Appendix J. Across multiple datasets, the simple transformer backbone equipped with the PErM strategy outperforms existing state-of-the-art (SOTA) models. The comparisons with UniTime and PatchTST are particularly noteworthy. As the methods that use the universal dataset setup, PErMformer outperforms UniTime by an average of 10% across various settings. Structurally, PatchTST is quite similar to PErM-None. When comparing PErMformer with PatchTST, we can also observe that the PErM strategy effectively enhances the model's prediction accuracy.

## 8 CONCLUSION

Owing to the spectral analysis from the perspective of adjacent embedding sequences, we observe that the embeddings generated by models trained solely with a transformer backbone exhibit a frequency collapse phenomenon, and we preliminarily demonstrate that this may lead to poor fitting for certain samples. To address this, we design the Post-Embedding reMapping strategy, introducing a remapping layer between the top layer of the transformer's backbone and the prediction head. This layer re-integrates the spectral representation of the embeddings and make the model easier to learn the high-frequency information. In future work, we aim to design more flexible and refined remapping functions to better model the frequency features, and further verify the effectiveness of this strategy on more time series foundational models trained with larger-scale datasets.

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

## A DETAILED PROOF OF EQUATION 1-3

*Assumption.* Suppose that all $\mathcal{E}(t)$ are stationary stochastic processes and uncorrelated with each other, and $\mathcal{E}_i$ is mapped to $\mathcal{Y}_i$ via a known deterministic matrix $W$.

*Derivation.* Given the stochastic processes $\mathcal{Y}^k(t)$ mapped from $\mathcal{E}^k(t)$ by $W$, consider the power spectral density $S_{\mathcal{Y}^k}(\omega)$ of $\mathcal{Y}^k(t)$ at frequency $\omega$, according to the definition of the power spectral density we have:

$$S_{\mathcal{Y}^k}(\omega) = \lim_{T \to +\infty} \left\{ \frac{1}{2T} \mathbb{E}| \int_{-T}^{T} \mathcal{Y}^k(t)\exp(-j\omega t)dt|^2 \right\} \tag{5}$$

As each $\mathcal{Y}^k(t) = \sum_{l=1}^{n} w_{k,l}\mathcal{E}^l(t)$, we have:

$$S_{\mathcal{Y}^k}(\omega) = \lim_{T \to +\infty} \left\{ \frac{1}{2T} \mathbb{E} \left| \sum_{l=1}^{n} w_{k,l} \int_{-T}^{T} \mathcal{E}^l(t)\exp(-j\omega t)dt \right|^2 \right\} \tag{6}$$

Since each stochastic process in $\mathcal{E}(t)$ is uncorrelated with each other, we expand the above equation and obtain:

$$S_{\mathcal{Y}^k}(\omega) = \lim_{T \to +\infty} \left\{ \frac{1}{2T} \sum_{l=1}^{n} w_{k,l}^2 \mathbb{E} \left( \int_{-T}^{T} \int_{-T}^{T} \mathcal{E}^l(t)\overline{\mathcal{E}^l(s)} \exp(-j\omega(t-s))dtds \right) \right\} \quad (7)$$

Using the autocorrelation function $R_X(t-s) = \mathbb{E}(X(t)\overline{X(s)})$, we have:

$$S_{\mathcal{Y}^k}(\omega) = \sum_{l=1}^{n} w_{k,l}^2 \left\{ \lim_{T \to +\infty} \frac{1}{2T} \int_{-T}^{T} \int_{-T}^{T} R_{\mathcal{E}^l}(t-s) \exp(-j\omega(t-s))dtds \right\} \quad (8)$$

where $R_{\mathcal{E}^l}(\cdot)$ denotes the autocorrelation function of the stationary stochastic process $\mathcal{E}^l(t)$. Let $\tau = t - s$, and we further get:

$$S_{\mathcal{Y}^k}(\omega) = \sum_{l=1}^{n} w_{k,l}^2 \left\{ \lim_{T \to +\infty} \int_{-2T}^{2T} (1 - \frac{|\tau|}{2T})R_{\mathcal{E}^l}(\tau) \exp(-j\omega\tau)d\tau \right\} \quad (9)$$

$$= \sum_{l=1}^{n} w_{k,l}^2 \int_{-\infty}^{+\infty} R_{\mathcal{E}^l}(\tau) \exp(-j\omega\tau)d\tau \quad (10)$$

$$= \sum_{l=1}^{n} w_{k,l}^2 S_{\mathcal{E}^l}(\omega) \quad (11)$$

where $S_{\mathcal{E}^l}(\omega)$ represents the power spectral density of the stochastic process signal $\mathcal{E}^l(t)$ at frequency $\omega$.

## B  SUPPLEMENTARY INFORMATION ON SPECTRAL ANALYSIS

### B.1  LIMITATIONS OF EXISTING SPECTRAL ANALYSIS ON TIME SERIES TASKS

Several recent works have focused on applying spectral analysis to motivate or improve their methods in time series topic (Fu et al., 2023; Zhou et al., 2022; Xu et al., 2024b). However, these methods perform spectral analysis merely on the raw time series to provide more information for the model. **Rather than conducting spectral analysis at the data level (input or output vector), we are more interested in performing spectral analysis at the embedding level to explore whether the model can effectively capture the diverse spectral information in the time series**, as shown in Fig. 10. By jointly analyzing the spectrum at the embedding level and the spectrum at the data level, we can preliminarily explore whether the model is capable of effectively capturing sufficiently rich frequency domain details. This can help us better understand the prediction results of the black-box models. For example, the loss of high-frequency information in the embeddings could result in the final predictions failing to accurately construct some detailed patterns of the time series.

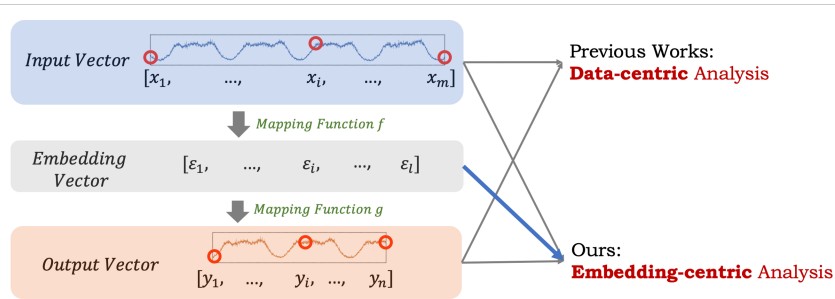

Figure 10: The difference between our approach and existing spectral analysis applied to time series tasks.

### B.2 Challenges in Embedding-level Spectral Analysis

The biggest challenge in implementing embedding-level spectral analysis lies in the fact that, unlike data-level spectral feature which can be easily captured in one sample, the embedding feature in one sample has little useful spectral information. After performing operations such as linear transformations, nonlinear activations, or even self-attention on the original time series sample $[x_1, \cdots, x_m]$, the generated embedding vectors $[e_1, \cdots, e_l]$ are unlikely to retain the periodic information of the original sequence. This can be easily proved, as for any two-layer neural network, we can construct an embedding that is different but functionally identical by swapping the positions of the intermediate layer's nodes and their connection matrices. As illustrated in Fig. 11, the transformed embeddings no longer exhibit meaningful spectral information. In other words, per-sample embedding spectral analysis lacks physical significance (i.e., disruption of temporal structure). Thus, thanks to the unique property in time series forecasting where the input vector and the input vector sequence share the temporal structure, we can perform the spectral analysis from the perspective of adjacent embedding sequence proposed in Sec. 3.

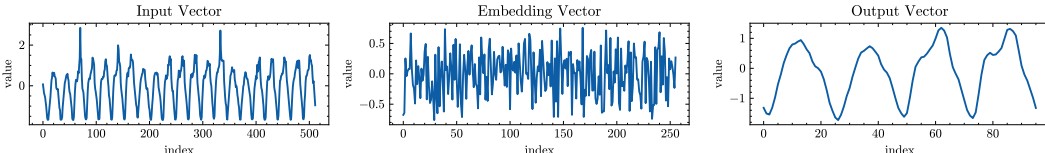

Figure 11: Per-sample Embedding cannot preserve the temporal structure of the raw time series.

## C Supplementary Analysis on Frequency Collapse

### C.1 Theoretical and Empirical Support for Frequency Collapse

In recent years, several works are dedicated to explore the behavior of deep neural networks (DNN) in a theoretical or empirical manner, especially under the framework of frequency analysis (Xu et al., 2024a; Molina et al., 2024; Luo et al., 2019; Rahaman et al., 2019a). They prove the existence of spectral bias at data-level: a DNN tends to learn a target function from low to high frequencies during training, which is also called Frequency Principle (F-principle). Moreover, Luo et al. (2019) theoretically proved that for the final training process of a DNN with parameter $\theta$ and L2 loss function, consider dividing the MSE loss function $L(\theta)$ into two parts, the high-frequency part $L^+(\theta)$ and low-frequency part $L^-(\theta)$, the proportion of gradients corresponding to high-frequency components, $\frac{\mathrm{d}L^+(\theta)/\mathrm{d}t}{\mathrm{d}L(\theta)/\mathrm{d}t}$, has a theoretical upper bound, and the proportion of gradients corresponding to low-frequency components, $\frac{\mathrm{d}L^-(\theta)/\mathrm{d}t}{\mathrm{d}L(\theta)/\mathrm{d}t}$, has a theoretical lower bound. This means compared to low-frequency signals, high-frequency signals are more difficult to learn, which is also confirmed by some empirical studies (Xu et al., 2024a). This can shed light on the embedding-level frequency collapse phenomenon even if the model is more complex and the perspective is not limited to data level.

### C.2 Quantitative Metric: High-Frequency Information Abundance

To more accurately measure the richness of high-frequency information in the embeddings generated by the model, we propose a quantitative metric based on adjacent embedding sequence frequency analysis: High-Frequency Information Abundance (HIA). Given certain model and specific test set, we test the model in a non-shuffle manner and extract the embedding $\mathcal{E}_i$ for each sample $\mathcal{X}_i$, and concatenating all embedding to generate the adjacent embedding sequence $\mathcal{E}_{seq}$. Then we perform spectral analysis for each sliding window $\mathcal{E}_{seq_{i,i+l}}$ with length $l$ of $\mathcal{E}_{seq}$. Specifically, assuming the embedding has $K$ features, for the $k$-th feature $\mathcal{E}_{seq_{i,i+l}}^k$, we obtain its spectrum and the amplitude $|X^k(\omega)|$ corresponding to each frequency $\omega$ via discrete Fourier transform (DFT).

The high-frequency spectrum $F^+[\mathcal{E}_{seq_{i,i+l}}]$ is defined as the set of normalized $|X(\omega)|$ where $\omega > C$ for all $K$ features, with $C$ representing the cutoff frequency (set to 20% in practice). The High-

Frequency Information Abundance of $\mathcal{E}_{seq_{i,i+l}}$ is defined as the ratio of the $2\text{-}Norm$ of its high-frequency spectrum and $\infty\text{-}Norm$ of its high-frequency spectrum:

$$HIA(\mathcal{E}_{seq_{i,i+l}}) = \frac{1}{K} \cdot \frac{\|F^+[\mathcal{E}_{seq_{i,i+l}}]\|_2}{\|F^+[\mathcal{E}_{seq_{i,i+l}}]\|_\infty},$$

and the overall High-Frequency Information Abundance is the average of all adjacent embedding sequences:

$$HIA(\mathcal{E}_{seq}) = \frac{1}{n-l} \sum_{i=1}^{n-l} HIA(\mathcal{E}_{seq_{i,i+l}}) = \frac{1}{(n-l) \cdot K} \sum_{i=1}^{n-l} \frac{\|F^+[\mathcal{E}_{seq_{i,i+l}}]\|_2}{\|F^+[\mathcal{E}_{seq_{i,i+l}}]\|_\infty},$$

where $n$ denotes the length of $\mathcal{E}_{seq}$ and $\|\cdot\|_p$ denotes the $p\text{-}Norm$ operation.

## D  SUPPLEMENTARY ANALYSIS ON PERM STRATEGY

### D.1  THE MECHANISM OF PERM FUNCTION

According to the theoretical support in C.1 that high-frequency signals are harder to fit, PErM strategy helps the model to learn the high-fequency information from a low-frequency perspective. To investigate how the PErM function influences the spectral distribution of embeddings, we construce a stochastic process using sinusoidal functions with multiple frequencies $w$, random phases $\phi$, and random amplitudes $a$: $X(t) = \sum_{i=1}^{3} a_i \sin(\omega_i t + \phi_i)$, and apply a simplified PErM function $\cos(0.1X(t))$ (a special case of RFF strategy). The spectrums of the stochastic process before and after the operation are shown in Fig. 12. According to the case, such a simplified remapping function can introduce more high frequency components and potentially result in harmonic components. Conversely, this implies that high-frequency signals in the final embedding can be represented by some lower-frequency signals before the PErM layer. According to F-principle, these lower-frequencies are easier to learn. In this way, the model can eventually model the signals more easily and precisely. RFF is a manually constructed mapping based on trigonometric functions and random factors. Since RFF is a more complex mapping, it allows the model to learn more robust mappings between frequencies from a greater number of channels. LPT function can be seen as a black-box, high-order non-linear mapping function that obtained from the "World Model" and have been empirically validated to have certain effects in some domains, thus we make an attempt to investigate if the mapping can alleviate frequency collapse.

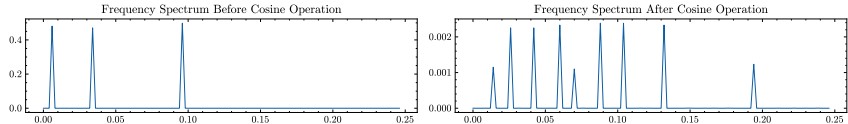

Figure 12: Spectrums of the stochastic process before and after the cosine operation.

### D.2  MORE EMPIRICAL EVIDENCE FOR THE EFFECTIVENESS OF PERM STRATEGY

We further validate the generality of the PErM strategy. Recent studies have shown that MLP-based models can achieve comparable or even better results than carefully designed Transformer-based models in time series forecasting tasks. In addition to verifying the PErM strategy on Transformer-based models, we explore its effectiveness for MLP-based time series foundation models. We conducte experiments for the MLP-based model under the same data settings as PErMformer, and employ the RFF mapping function. Since the representation capacity of MLP-based model is limited and it can converge more easily to optimal values, the results in Table 4 show that the PErM strategy also brings significant performance improvements to MLP-based models on those datasets with higher sample rate (ETTm series and Weather) which might include more high-frequency features.

Based on the quantitative metric proposed in Sec. C.2, we measure the changes in high-frequency information in the embeddings of MLP-based and Transformer-based models with and without the PErM strategy. From Fig. 13, we can conclude that PErM strategy can effectively enhance the high-frequency representation in the embedding, which theoretically raise the upper limit of the model's ability to capture high-frequency information in the raw time series.

Table 4: Forecasting results of the three models on seven datasets in MSE with MLP backbone. The best result is highlighted in bold. Imp. denotes the performance improvement comparing with the method without the PErM strategy, and "-" denotes the improvement is not significant.

|  | ECL | ETTh1 | ETTh2 | ETTm1 | ETTm2 | Traffic | Weather |
|---|---|---|---|---|---|---|---|
| MLP | 0.136 | 0.322 | 0.296 | 0.339 | 0.203 | 0.377 | 0.176 |
| PErM-MLP | 0.135 | 0.319 | 0.293 | 0.300 | 0.189 | 0.375 | 0.162 |
| Imp. | - | - | - | 11.8% ↑ | 6.9% ↑ | - | 8.0% ↑ |

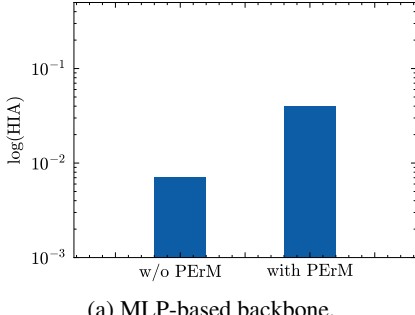

(a) MLP-based backbone.

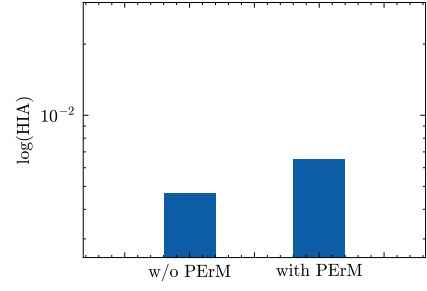

(b) Transformer-based backbone.

Figure 13: HIA of MLP-based and Transformer-based models with and without the PErM strategy.

# E    DETAILS OF DATASETS

The number of time steps, number of channels, sample rate, and the description of each dataset are listed in Table 5. All datasets are divided into training, validation, and test sets with a 7:1:2 ratio.

Table 5: The detailed information of datasets.

| Datasets | Time steps | Channels | Sample rate | Description |
|---|---|---|---|---|
| ECL (2021) | 26304 | 321 | 1 hour | Electricity consumption data of 321 clients |
| ETTh1 (2021) | 17420 | 7 | 1 hour | 7 factors of electricity transformer |
| ETTh2 (2021) | 17420 | 7 | 1 hour | 7 factors of electricity transformer |
| ETTm1 (2021) | 69680 | 7 | 15 min | 7 factors of electricity transformer |
| ETTm2 (2021) | 69680 | 7 | 15 min | 7 factors of electricity transformer |
| Traffic (2021) | 17544 | 862 | 1 hour | Road occupancy rates measured by 862 sensors |
| Weather (2021) | 52696 | 21 | 10 min | 21 meteorological factors |

# F    IMPLEMENTATION DETAILS

## F.1    DETAILS OF PERMFORMER AND THE SPECTROGRAM

All three variants *PErM-FPT*, *PErM-RFF*, and *PErM-None* use the same parameter set. Each model contains 12 transformer blocks. We use the ADAM (Kingma, 2014) optimizer and the learning rate is set to $2 \times 10^{-4}$. We use the L2 loss for model optimization. The dropout rate is set to 0.1. The input series is split into non-overlapped patches whose length is 16, so there are 32 patches for each sample (the lookback window is 512). For the transformer block, the $n\_head$ is set to 8, and the dimension of the heads are set to 64. The dimension of the hidden state ($d\_model$) is set to 256, thus the dimension of the embedding sent into the prediction head is $patches \times d\_model = 8192$. This also represents the length of the Feature Index axis in the spectrogram.

### F.2 DETAILS OF THE BASELINES

We reproduced all the results of baselines. Specifically, the code of Time-LLM comes from `https://github.com/KimMeen/Time-LLM`. The codes of DLinear, FEDformer and Informer come from `https://github.com/cure-lab/LTSF-Linear`. Others come from the TSlib repository (Wang et al. (2024b), `https://github.com/thuml/Time-Series-Library`). The configuration of the lookback window follows the setting in the original paper.

## G PERFORMANCE UNDER DIFFERENT LPT SETTINGS

For the PErM-LPT method, we experiment with using different layers from various pre-trained large language models as the remapping layer. Through our experiments, we find that using the middle and top layers of large language models tends to yield better results. We hypothesize that this may be due to the deeper layers of large language models having more abstract semantic extraction capabilities. Although it is difficult to effectively interpret what the mapping relationships of a specific layer in a large language model represent, we find that the mapping relationships represented by smaller LLM may lack a certain degree of generalization (e.g., GPT-2).

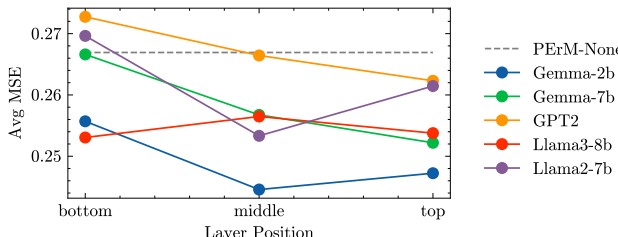

Figure 14: The performance of using different LLM layers as the PErM function.

## H MORE COMPARISON OF THE THREE MODELS

We validate the enhancement of high-frequency information by the PErM strategy across multiple datasets. We visualize several prediction samples and present the spectrograms of their adjacent embedding sequences as follows.

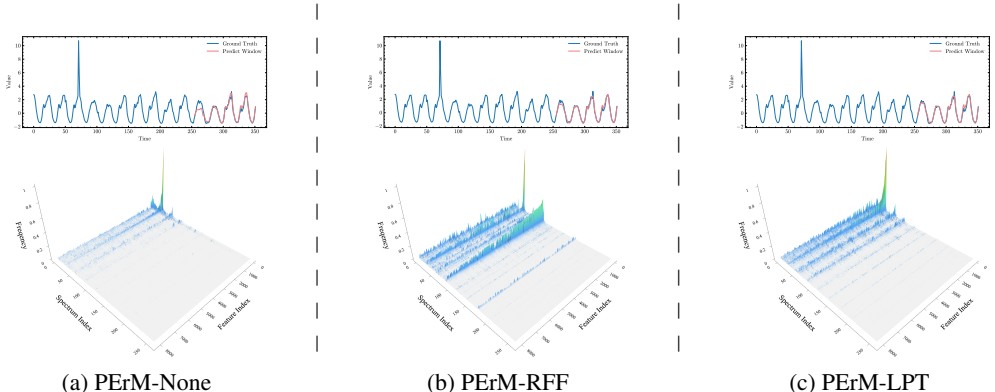

| (a) PErM-None | (b) PErM-RFF | (c) PErM-LPT |
|:---:|:---:|:---:|

Figure 15: The prediction results and the corresponding spectrograms of the embedding sequences of the three models on the same lookback window on dataset Traffic.

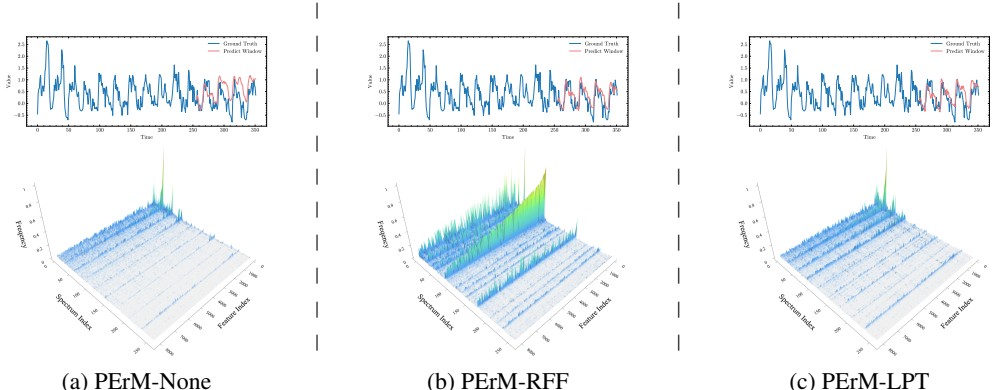

(a) PErM-None                    (b) PErM-RFF                    (c) PErM-LPT

Figure 16: The prediction results and the corresponding spectrograms of the embedding sequences of the three models on the same lookback window on dataset ETTh1.

## I    CASES OF PREDICTION COMPARED WITH SOTA METHODS

We visualize the prediction performance of our method compared to several state-of-the-art methods across multiple datasets (Fig. 17, Fig. 18, Fig. 19, and Fig. 20).

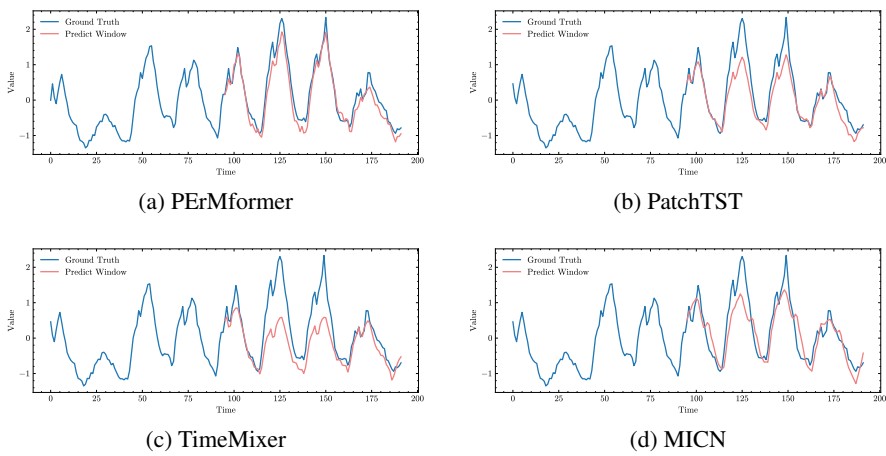

(a) PErMformer                                      (b) PatchTST

(c) TimeMixer                                        (d) MICN

Figure 17: Visualization of the prediction on the ECL dataset.

## J    FULL RESULTS

The full results that contain all the baselines are listed in Table 6 . Due to computational resource constraints, we reproduce the performance of the TimeLLM only on the datasets reported in its original paper (ETT series).

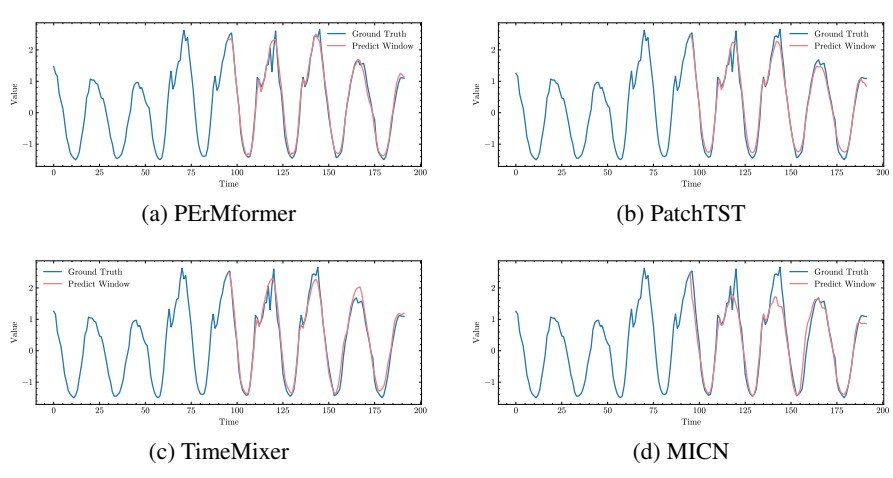

Figure 18: Visualization of the prediction on the Traffic dataset.

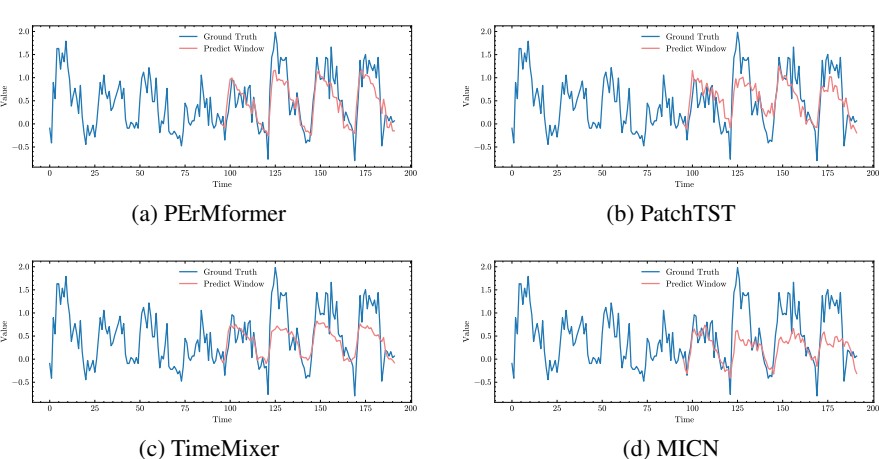

Figure 19: Visualization of the prediction on the ETTh1 dataset.

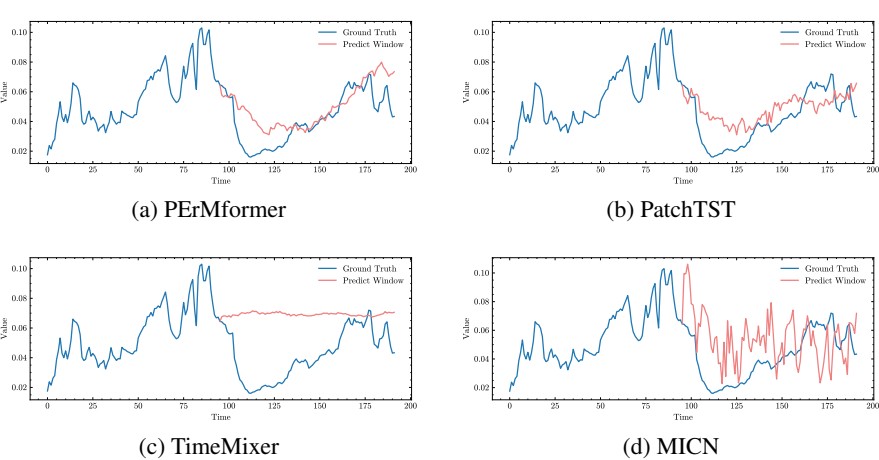

Figure 20: Visualization of the prediction on the Weather dataset.

Table 6: The full results of all baselines. The best results are highlighted in red, and the second-best results are highlighted in blue. '-' denotes that the experimental result cannot be obtained within the limited time constraint (2 days per GPU).

| Dataset | Horizon | PEMformer MSE | PEMformer MAE | FITS MSE | FITS MAE | iTransformer MSE | iTransformer MAE | PatchTST MSE | PatchTST MAE | UniTime MSE | UniTime MAE | TSMixer MSE | TSMixer MAE | TimeMixer MSE | TimeMixer MAE | DLinear MSE | DLinear MAE | MICN MSE | MICN MAE | Crossformer MSE | Crossformer MAE | TimesNet MSE | TimesNet MAE | TimeLLM MSE | TimeLLM MAE | Nonstationary MSE | Nonstationary MAE | FEDformer MSE | FEDformer MAE | Pyraformer MSE | Pyraformer MAE | Autoformer MSE | Autoformer MAE | Informer MSE | Informer MAE |
|---|---|---|---|---|---|---|---|---|---|---|---|---|---|---|---|---|---|---|---|---|---|---|---|---|---|---|---|---|---|---|---|---|---|---|---|
| ECL | 96 | 0.131 | 0.228 | 0.144 | 0.242 | 0.148 | 0.239 | 0.136 | 0.239 | 0.183 | 0.274 | 0.204 | 0.308 | 0.156 | 0.247 | 0.14 | 0.237 | 0.165 | 0.276 | 0.151 | 0.253 | 0.168 | 0.272 | - | - | 0.169 | 0.269 | 0.187 | 0.302 | 0.278 | 0.374 | 0.21 | 0.324 | 0.323 | 0.409 |
| ECL | 192 | 0.149 | 0.244 | 0.164 | 0.267 | 0.167 | 0.258 | 0.153 | 0.28 | 0.189 | 0.28 | 0.218 | 0.329 | 0.17 | 0.261 | 0.154 | 0.25 | 0.176 | 0.288 | 0.159 | 0.259 | 0.187 | 0.289 | - | - | 0.183 | 0.283 | 0.195 | 0.31 | 0.294 | 0.389 | 0.225 | 0.334 | 0.352 | 0.434 |
| ECL | 336 | 0.166 | 0.263 | 0.222 | 0.339 | 0.178 | 0.271 | 0.248 | 0.264 | 0.204 | 0.294 | 0.239 | 0.35 | 0.185 | 0.278 | 0.169 | 0.268 | 0.185 | 0.297 | 0.203 | 0.283 | 0.203 | 0.304 | - | - | 0.204 | 0.307 | 0.212 | 0.327 | 0.292 | 0.388 | 0.247 | 0.352 | 0.348 | 0.432 |
| ECL | 720 | 0.218 | 0.312 | 0.3 | 0.3 | 0.209 | 0.298 | 0.246 | 0.328 | 0.242 | 0.325 | 0.272 | 0.373 | 0.242 | 0.312 | 0.204 | 0.3 | 0.231 | 0.318 | 0.254 | 0.318 | 0.254 | 0.343 | - | - | 0.219 | 0.318 | 0.25 | 0.358 | 0.297 | 0.386 | 0.277 | 0.352 | 0.405 | 0.463 |
| ETTh1 | 96 | 0.317 | 0.371 | 0.334 | 0.376 | 0.337 | 0.383 | 0.365 | 0.409 | 0.343 | 0.392 | 0.414 | 0.46 | 0.376 | 0.404 | 0.352 | 0.402 | 0.337 | 0.396 | 0.382 | 0.441 | 0.347 | 0.395 | 0.348 | 0.395 | 0.429 | 0.46 | 0.376 | 0.415 | 0.601 | 0.598 | 0.47 | 0.457 | 0.746 | 0.669 |
| ETTh1 | 192 | 0.356 | 0.394 | 0.379 | 0.408 | 0.383 | 0.412 | 0.409 | 0.441 | 0.385 | 0.419 | 0.516 | 0.531 | 0.374 | 0.404 | 0.391 | 0.431 | 0.396 | 0.447 | 0.462 | 0.493 | 0.401 | 0.43 | 0.365 | 0.408 | 0.46 | 0.558 | 0.423 | 0.446 | 0.687 | 0.641 | 0.464 | 0.468 | 0.951 | 0.778 |
| ETTh1 | 336 | 0.372 | 0.407 | 0.397 | 0.414 | 0.412 | 0.439 | 0.46 | 0.473 | 0.42 | 0.44 | 0.597 | 0.597 | 0.427 | 0.431 | 0.407 | 0.452 | 0.407 | 0.452 | 0.546 | 0.549 | 0.431 | 0.445 | 0.414 | 0.449 | 0.665 | 0.599 | 0.459 | 0.466 | 0.741 | 0.679 | 0.494 | 0.482 | 1.073 | 0.815 |
| ETTh1 | 720 | 0.373 | 0.425 | 0.388 | 0.433 | 0.443 | 0.465 | 0.626 | 0.559 | 0.454 | 0.454 | 0.706 | 0.665 | 0.415 | 0.438 | 0.461 | 0.498 | 0.524 | 0.535 | 0.948 | 0.783 | 0.438 | 0.464 | 0.444 | - | 0.746 | 0.656 | 0.406 | 0.457 | 0.975 | 0.799 | 0.49 | - | 1.113 | 0.832 |
| ETTh2 | 96 | 0.299 | 0.35 | 0.3 | 0.347 | 0.31 | 0.354 | 0.33 | 0.379 | 0.326 | 0.369 | 0.823 | 0.664 | 0.305 | 0.351 | 0.27 | 0.346 | 0.362 | 0.406 | 0.716 | 0.585 | 0.343 | 0.371 | 0.28 | 0.344 | 0.354 | 0.383 | 0.34 | 0.384 | 0.444 | 0.465 | 0.443 | 0.475 | 2.845 | 1.336 |
| ETTh2 | 192 | 0.381 | 0.396 | 0.374 | 0.394 | 0.39 | 0.402 | 0.387 | 0.421 | 0.377 | 0.395 | 2.831 | 1.461 | 0.382 | 0.397 | 0.394 | 0.421 | 0.492 | 0.492 | 1.652 | 0.934 | 0.404 | 0.41 | 0.39 | 0.406 | 0.547 | 0.491 | 0.431 | 0.44 | 5.794 | 1.921 | 0.472 | 0.482 | 5.324 | 1.886 |
| ETTh2 | 336 | 0.416 | 0.427 | 0.401 | 0.42 | 0.434 | 0.437 | 0.451 | 0.459 | 0.419 | 0.429 | 1.5 | 1.5 | 0.434 | 0.436 | 0.507 | 0.493 | 0.603 | 0.538 | 3.483 | 1.475 | 0.462 | 0.455 | 0.381 | 0.413 | 0.651 | 0.544 | 0.501 | 0.494 | 5.117 | 1.856 | 0.488 | 0.497 | 4.444 | 1.722 |
| ETTh2 | 720 | 0.403 | 0.435 | 0.416 | 0.437 | 0.433 | 0.446 | 0.46 | 0.473 | 0.42 | 0.44 | 2.973 | 1.302 | 0.415 | 0.438 | 0.895 | 0.681 | 0.524 | 0.535 | 2.524 | 1.301 | 0.471 | 0.473 | 0.446 | 0.461 | 0.731 | 0.584 | 0.477 | 0.484 | 3.341 | 1.552 | 0.52 | 0.51 | 4.38 | 1.773 |
| ETTm1 | 96 | 0.275 | 0.345 | 0.266 | 0.325 | 0.337 | 0.383 | 0.276 | 0.332 | 0.326 | 0.369 | 0.369 | 0.423 | 0.278 | 0.339 | 0.27 | 0.346 | 0.346 | 0.346 | 0.358 | 0.413 | 0.33 | 0.37 | 0.28 | 0.344 | 0.354 | 0.383 | 0.364 | 0.413 | 0.444 | 0.465 | 0.509 | 0.475 | 0.443 | 0.474 |
| ETTm1 | 192 | 0.323 | 0.375 | 0.3 | 0.347 | 0.368 | 0.412 | 0.316 | 0.364 | 0.366 | 0.39 | 0.393 | 0.446 | 0.318 | 0.361 | 0.306 | 0.373 | 0.373 | 0.373 | 0.371 | 0.438 | 0.372 | 0.393 | 0.311 | 0.363 | 0.448 | 0.438 | 0.406 | 0.455 | 0.519 | 0.539 | 0.554 | 0.552 | 0.443 | 0.474 |
| ETTm1 | 336 | 0.336 | 0.386 | 0.333 | 0.368 | 0.392 | 0.439 | 0.344 | 0.381 | 0.396 | 0.411 | 0.491 | 0.491 | 0.337 | 0.381 | 0.352 | 0.408 | 0.408 | 0.408 | 0.372 | 0.527 | 0.404 | 0.404 | 0.337 | 0.379 | 0.454 | 0.451 | 0.444 | 0.458 | 0.616 | 0.599 | 0.548 | 0.501 | 0.699 | 0.83 |
| ETTm1 | 720 | 0.403 | 0.435 | 0.422 | 0.432 | 0.425 | 0.431 | 0.403 | 0.422 | 0.452 | 0.443 | 0.583 | 0.583 | 0.399 | 0.42 | 0.392 | 0.433 | 0.433 | 0.433 | 0.46 | 0.624 | 0.453 | 0.453 | 0.386 | 0.409 | 0.554 | 0.511 | 0.531 | 0.499 | 0.688 | 0.688 | 0.498 | 0.498 | 0.781 | 1.021 |
| ETTm2 | 96 | 0.191 | 0.27 | 0.181 | 0.262 | 0.191 | 0.272 | 0.188 | 0.272 | 0.179 | 0.266 | 0.257 | 0.361 | 0.183 | 0.261 | 0.192 | 0.262 | 0.192 | 0.285 | 0.358 | 0.4 | 0.19 | 0.266 | 0.191 | 0.276 | 0.304 | 0.34 | 0.189 | 0.282 | 0.345 | 0.437 | 0.257 | 0.323 | 0.345 | 0.432 |
| ETTm2 | 192 | 0.25 | 0.313 | 0.244 | 0.301 | 0.264 | 0.318 | 0.254 | 0.317 | 0.246 | 0.309 | 0.565 | 0.59 | 0.251 | 0.305 | 0.247 | 0.314 | 0.306 | 0.369 | 0.531 | 0.521 | 0.266 | 0.313 | 0.242 | 0.31 | 0.45 | 0.415 | 0.255 | 0.324 | 1.247 | 0.854 | 0.284 | 0.339 | 0.758 | 0.669 |
| ETTm2 | 336 | 0.289 | 0.338 | 0.301 | 0.338 | 0.318 | 0.358 | 0.295 | 0.341 | 0.309 | 0.35 | 0.59 | 0.751 | 0.311 | 0.348 | 0.3 | 0.355 | 0.37 | 0.408 | 0.531 | 0.521 | 0.332 | 0.351 | 0.242 | 0.342 | 0.45 | 0.415 | 0.327 | 0.365 | 0.846 | 0.696 | 0.339 | 0.375 | 0.851 | 0.669 |
| ETTm2 | 720 | 0.432 | 0.44 | 0.422 | 0.432 | 0.425 | 0.431 | 0.403 | 0.422 | 0.452 | 0.443 | 2.463 | 1.302 | 0.399 | 0.42 | 0.392 | 0.483 | 0.521 | 0.493 | 4.365 | 1.438 | 0.435 | 0.411 | 0.418 | 0.421 | 0.562 | 0.475 | 0.439 | 0.428 | 3.509 | 1.339 | 0.515 | 0.51 | 4.824 | 1.624 |
| Traffic | 96 | 0.361 | 0.256 | 0.411 | 0.281 | 0.392 | 0.268 | 0.368 | 0.257 | 0.477 | 0.309 | 0.531 | 0.358 | 0.476 | 0.297 | 0.413 | 0.287 | 0.522 | 0.313 | 0.533 | 0.274 | 0.589 | 0.315 | - | - | 0.625 | 0.349 | 0.573 | 0.366 | 0.693 | 0.399 | 0.67 | 0.401 | 0.749 | 0.418 |
| Traffic | 192 | 0.377 | 0.262 | 0.423 | 0.286 | 0.413 | 0.277 | 0.385 | 0.265 | 0.479 | 0.31 | 0.563 | 0.385 | 0.508 | 0.301 | 0.424 | 0.29 | 0.537 | 0.318 | 0.58 | 0.296 | 0.617 | 0.326 | - | - | 0.642 | 0.354 | 0.608 | 0.378 | 0.683 | 0.389 | 0.646 | 0.413 | 0.77 | 0.434 |
| Traffic | 336 | 0.392 | 0.272 | 0.434 | 0.291 | 0.425 | 0.283 | 0.396 | 0.272 | 0.494 | 0.316 | 0.58 | 0.393 | 0.508 | 0.308 | 0.438 | 0.299 | 0.55 | 0.323 | 0.571 | 0.293 | 0.635 | 0.338 | - | - | 0.645 | 0.355 | 0.622 | 0.381 | 0.691 | 0.391 | 0.61 | 0.382 | 0.86 | 0.485 |
| Traffic | 720 | 0.453 | 0.316 | 0.463 | 0.308 | 0.458 | 0.3 | 0.439 | 0.3 | 0.528 | 0.333 | 0.619 | 0.418 | 0.547 | 0.323 | 0.466 | 0.316 | 0.576 | 0.334 | 0.607 | 0.301 | 0.659 | 0.349 | - | - | 0.662 | 0.36 | 0.631 | 0.382 | 0.73 | 0.411 | 0.661 | 0.416 | 1.053 | 0.592 |
| Weather | 96 | 0.156 | 0.21 | 0.146 | 0.198 | 0.178 | 0.219 | 0.158 | 0.214 | 0.172 | 0.215 | 0.18 | 0.252 | 0.162 | 0.208 | 0.174 | 0.233 | 0.193 | 0.25 | 0.174 | 0.233 | 0.169 | 0.219 | 0.18 | - | 0.18 | 0.23 | 0.217 | 0.296 | 0.199 | 0.281 | 0.275 | 0.341 | 0.352 | 0.423 |
| Weather | 192 | 0.206 | 0.257 | 0.19 | 0.24 | 0.226 | 0.259 | 0.213 | 0.266 | 0.219 | 0.255 | 0.218 | 0.287 | 0.208 | 0.251 | 0.218 | 0.278 | 0.24 | 0.301 | 0.232 | 0.301 | 0.225 | 0.265 | - | - | 0.237 | 0.28 | 0.289 | 0.348 | 0.221 | 0.307 | 0.309 | 0.366 | 0.59 | 0.53 |
| Weather | 336 | 0.262 | 0.296 | 0.242 | 0.279 | 0.282 | 0.3 | 0.248 | 0.264 | 0.274 | 0.295 | 0.261 | 0.321 | 0.264 | 0.293 | 0.282 | 0.314 | 0.282 | 0.331 | 0.282 | 0.34 | 0.282 | 0.304 | - | - | 0.31 | 0.329 | 0.357 | 0.398 | 0.29 | 0.307 | 0.401 | 0.424 | 0.805 | 0.648 |
| Weather | 720 | 0.327 | 0.344 | 0.322 | 0.338 | 0.358 | 0.35 | 0.32 | 0.338 | 0.352 | 0.345 | 0.318 | 0.363 | 0.345 | 0.346 | 0.35 | 0.374 | 0.35 | 0.387 | 0.372 | 0.412 | 0.359 | 0.354 | - | - | 0.418 | 0.394 | 0.407 | 0.422 | 0.399 | 0.414 | 0.426 | 0.429 | 1.142 | 0.782 |
| 1st | | 13 | 13 | 4 | 9 | 0 | 0 | 1 | 0 | 1 | 1 | 1 | 0 | 0 | 1 | 6 | 1 | 0 | 0 | 0 | 0 | 0 | 0 | 2 | 1 | 0 | 0 | 0 | 0 | 0 | 0 | 0 | 0 | 0 | 0 |

