# OpenReview forum: "Dealing with Frequency Collapse in Time Series Embeddings by Post-Embedding reMapping"
_ICLR.cc/2025/Conference — ICLR 2025 Conference Withdrawn Submission_

### Official Review · Reviewer_GVwq · 2024-10-30

**Soundness:** 2
**Presentation:** 3
**Contribution:** 2
**Rating:** 5
**Confidence:** 4

**Summary:**

The paper presents a novel approach for improving time series forecasting using transformer-based models, addressing a phenomenon termed "frequency collapse" where high-frequency signals critical for accurate prediction are lost in the embeddings. This underfitting problem particularly affects extreme value regions, leading to significant deviations between predicted values and actual time series patterns. To counter this, the authors introduce the Post-Embedding ReMapping (PErM) strategy, incorporating fixed non-linear remapping functions to enhance high-frequency representation in embeddings without needing additional training.

**Strengths:**

1. Introduces the concept of frequency collapse in time series models, offering fresh insight into the limitations of transformer-based forecasting.
2. The proposed PErM layer is simple and plug-and-play, without increasing much complexity of the module.
3. Thorough comparisons with other advanced methods underscore the robustness of the PErM approach.

**Weaknesses:**

1. The reason why the method can avoid frequency collapse is not presented clearly, especially for LPT. Also, the author could present some connections between RFF and LPT.
2. Some of the compared results are not consistent with previous research. For example, the PatchTST results on Traffic/Electricity are reported lower than the original paper[1]. The author should explain this inconsistency and make more fair comparisons.
3. Low frequency principle often leads to better generalization in deep learning [2]. And there are some methods that deliberately cutoff high-frequency information to achieve better performance [3]. The author should explain the connection and contradiction of these papers.
4. The paper's analysis and solution are limited to the patch-based methods. Whether the conclusions are also applicable to other methods is not clear.
5. Some of the existing research has pointed out that channel independence leads to smooth prediction [4]. Therefore, the author should include a discussion on the research and explore the correlation between frequency collapse and channel independence.

[1] Yuqi Nie, Nam H. Nguyen, Phanwadee Sinthong, Jayant Kalagnanam: A Time Series is Worth 64 Words: Long-term Forecasting with Transformers. ICLR 2023

[2] Zhi-Qin John Xu, Yaoyu Zhang, Tao Luo, Yanyang Xiao, Zheng Ma: Frequency Principle: Fourier Analysis Sheds Light on Deep Neural Networks. CoRR abs/1901.06523 (2019)

[3] Zhijian Xu, Ailing Zeng, Qiang Xu: FITS: Modeling Time Series with 10k Parameters. ICLR 2024

[4]  Lu Han, Han-Jia Ye, De-Chuan Zhan: The Capacity and Robustness Trade-Off: Revisiting the Channel Independent Strategy for Multivariate Time Series Forecasting. IEEE Trans. Knowl. Data Eng. 36(11): 7129-7142 (2024)

**Questions:**

See weaknesses.

---

> ### Author Response · Authors · 2024-11-21
> **Response to your concerns**
>
> Thanks for your comprehensive and valuable review. We will address your concerns one by one.
>
> ## \# Weakness 1:
> We appreciate your valuable suggestions. To address your concerns, we added a section **"Supplementary Analysis on PErM Strategy"** as Appendix D, and it provides further explanations on the mechanism of PErM.
>
> In brief, **we show that high-frequency signals in the final embedding can be represented by lower-frequency signals before the PErM layer**. According to F-principle[1], these lower-frequencies are easier to learn. In this way, the model can eventually model the signals more precisely. Compared to the manually constructed function RFF, **LPT function can be seen as a black-box, high-order non-linear mapping function obtained from the "World Model"**, and according to the empirical study of our experiments and previous works, this highly abstract and non-interpretable function can achieve results comparable to or even surpass RFF.
>
> ## \# Weakness 2:
> That's quite a meaningful question. Our experiments for PatchTST is based on the TSlib repo, and the code has been validated by the author of PatchTST. We follow the hyperparameters as their original paper, and repeat the experiments three times using different random seeds. So one possible reason is that insufficient model robustness leads to larger random errors.
>
> ## \# Weakness 3:
> Thanks for your insightful opinion. For tasks that do not require very high resolution, such as classification tasks, the low-frequency principle may indeed provide some benefits on generalization. However, for generative tasks, the lack of high-frequency modeling capabilities can result in poor generation performance. In image tasks, this means a loss of details and blurriness, while **in time series forecasting tasks, this means pattern shifts and over-smoothing on extreme values**.
>
> Actually, for time series models like FITS, the purpose of cutting off high frequencies is to reduce the number of parameters and increase efficiency in the training and inference phase, and we empirically find that this operation can lead to underfitting.
>
> Specifically, we apply the PErM function RFF to FITS to enhance its ability to represent high-frequency signals, and from the below results, the RFF function can improve the overall performance. This indicates that high-frequency information plays an important role in prediction.
>
> |   FITS Variants   | vanilla | with PErM   |
> | ---- | ---- | ---- |
> | Avg. MSE |   0.254   |   0.248  |
>
> ## \# Weakness 4:
> That's an interesting point. We further explored the effectiveness of the PErM strategy on the MLP-based model which is not the patch-based model. Different from FITS which trains one model for each dataset, this model shares the setting with PErMformer, i.e., training one model for all datasets. As shown in the table, the PErM strategy can bring significant performance improvement:
>
> |   MLP-backbone Variants   | w/o PErM | with PErM |
> | ---- | ---- | ---- |
> | Avg. MSE |   0.264   |  0.253  |
>
> Also, if you are interested in more information about this experiment, please refer to the section **"More Empirical Evidence for the Effectiveness of PErM Strategy"** (Appendix D.2) in our revised PDF.
>
> ## \# Weakness 5:
> The channel setting is indeed a hot topic in the time series domain. Under the channel independence setting, the model only needs to focus on the temporal structure of the time series, making it relatively easier to conduct frequency analysis specifically targeting the temporal structure. This is the primary reason we chose this setting in our current work.
>
> In contrast, under the channel-mixing setting, the model also needs to capture the dependency between metrics. In this case, frequency analysis must take more factors like inter-metric dependency into consideration, and the analysis framework would likely become more complex. This will be the focus of our future efforts.
>
> ***If you have any other questions or have some insights to share, we welcome further discussion with you.***
>
> [1] Luo, Tao, et al. "Theory of the frequency principle for general deep neural networks." arXiv preprint arXiv:1906.09235 (2019).

---

> > ### Comment · Reviewer_GVwq · 2024-11-25
> > **Reply to authors' rebuttal**
> >
> > Thank you for your response. I have carefully reviewed your rebuttal. While some of my concerns have been addressed, others remain unresolved:
> >
> > 1. The connection between RFF and LPT has not been clearly explained, which undermines the overall logical coherence of the paper.
> > 2. PatchTST has an official reproducible code. The results reproduced using third-party code differ from those in the original paper. This raises questions about the reliability of these results.
> > 3. The analysis in the paper is limited to the channel-independent patch-based Transformer approach, i.e. PatchTST, and the generalizability of the conclusions has not been demonstrated.
> >
> > Given these points, I believe the paper still needs improvement. Therefore, I have decided to maintain my score.

---

### Official Review · Reviewer_nD11 · 2024-11-03

**Soundness:** 1
**Presentation:** 1
**Contribution:** 1
**Rating:** 3
**Confidence:** 4

**Summary:**

This paper introduces a method to improve the performance of transformer-based time series forecasting methods by using a fixed transformation to supposedly enhance the model's ability to handle high-frequency content in input signals. However, as detailed below, there are several issues in the presentation, soundness, and evaluation of the approach.

**Strengths:**

* Studying the inductive biases of modern sequence modeling architectures is an important topic that deserves more attention, and examining the spectrum of the extracted representations is a principled approach to this problem.
* The fact that transformer encoders might lose fine-grained information is also being explored in the literature (particularly in NLP [1]), and this aspect definitely deserves more attention in the context of time series.

**Weaknesses:**

### Main issues

There are several issues.

* **Significance and soundness** Sections 3 and 4 present many obvious facts and report them as novel findings, offering only anecdotal evidence that can hardly support any claim. In particular, the paper presents as "novel" the idea of performing a Fourier transform on representations obtained by feeding the data to the model with a sliding window approach. The entirety of Section 3 reiterates this by showing something as obvious as a matrix with shifted indices in each column. Note also that embeddings produced by any autoregressive sequence model (e.g., from RNN or TCN architectures) preserve the temporal structure of the input; the same can be said for transformer architectures tailored to processing sequences (if the author think that is not the case, this would need to be showed). The theoretical analysis at the end of the section is, again, not particularly informative as it's a simple rehashing of linear algebra facts under overly simplistic assumptions. Section 4 merely shows some qualitative anecdotal results on a single dataset with a single model. Figure 6 once again shows something as obvious as the loss of high-frequency content after applying a low-pass filter to representations extracted by the model. I’m not disputing that transformers may suffer from oversmoothing (other recent work suggests they do, e.g., [1]), but rather that the analysis in this paper lacks depth and significance. Lastly, the adoption of random features and pre-trained LLM layers as fixed transformations should be further justified (see also comments on empirical evaluation).

* **Presentation** The related work section is incomplete, and the paper lacks context on similar studies in the literature. Frequency response analysis is a well-studied area in time series analysis and signal processing (e.g., see [2] for a classic reference) and is commonly used for designing filters and models. The paper should do a much better job positioning its results within the existing literature. Concerning recent ML research, filter analysis and frequency response have been studied, for example, in the context of structured sequence models [3]. Oversquashing in transformers has been studied in [1]. See also minor comments regarding the correctness of several statements.

* **Empirical evaluation** In many of the benchmarks used in the evaluation, a simple autoregressive linear model trained with ordinary least squares can achieve results far better than those reported in the paper, as shown by [4]. Additionally, [5] showed that in these benchmarks, even removing transformer and pre-trained blocks entirely from similar architectures achieves comparable performance to the full model, further hinting at limitations in the presented empirical analysis. The empirical analysis should rather focus on assessing the presence of the discussed phenomena across different relevant baselines and reference architectures. Further, all results should at least be reported with standard deviations concerning different model initializations.

Given the above, I believe the paper would require a complete rework and a much more in-depth analysis. As such, I cannot recommend acceptance, but I do think this direction has potential.

### Minor comments

* Why is the positional encoding deliberately removed from the adopted model? (lines 301-304)
* Several statements lack clarity or justification. Some examples:
    - "The model’s forward process disrupts the internal temporal structure of the samples, thus the temporal information within each embedding is disordered. This means that performing spectral analysis directly on the embeddings is meaningless." Which model? What is the reference Transformer class? What are the assumptions here?
    - "[...] the sliding window sampling method allows the sample sequence to directly reflect the internal temporal structure of each sample. Thanks to this finding [...]" This is not a finding; it is a well-known basic fact.
    - "Fortunately, we notice a notable characteristic of time series tasks is the strong correlation between adjacent samples, which is different from other tasks." See the above.
    - "We further prove the absence of high-frequency signals in the embedding sequence." As motivated above, I believe this claim is not adequately justified.
    - "Since time series data inherently exhibits autocorrelation, employing positional embedding may disrupt this autocorrelation." How? Why?

#### References

[1] Barbero et al., "Transformers need glasses! Information over-squashing in language tasks" NeurIPS 2024\
[2] Strang and Nguyen, "Wavelets and filter banks" 1996\
[3] Fu et al., "Simple Hardware-Efficient Long Convolutions for Sequence Modeling" ICML 2023\
[4] Toner et al., "An Analysis of Linear Time Series Forecasting Models." ICML 2024\
[5] Tan et al. "Are language models actually useful for time series forecasting?" NeurIPS 2024

**Questions:**

See weaknesses.

---

> ### Author Response · Authors · 2024-11-21
> **Response to your concerns (Part 1/2)**
>
> Thanks for your comprehensive and helpful review. Your suggestions and concerns remind us that **it is necessary to review the relevant work on frequency domain analysis in time series forecasting along with their limitations, and to provide a detailed explanation of the challenges we encountered when we try to perform spectral analysis on embeddings**.
>
> Thus we add the Section **"Supplementary Information on Spectral Analysis"** in Appendix of the revised PDF (**Appendix B**). We hope this will address some of your concerns regarding the "Significance and Soundness" and "Minor comments" section.
>
> Based on the supplementary section, we will address each of your concerns one by one.
>
> ##  \# Weakness in *Significance and soundness*:
>
>  To summary, the significance of our analysis lies in 1) making embedding-level spectral analysis possible by converting sample-level analysis to sequence-level analysis (Section 3), and 2) mitigating the gap between the sequence-level analysis result "frequency collapse" and the sample-level prediciton result "underfitting" via both theoretical and empirical analysis (Section 3 and 4).
>
>  Here are the detailed responses for each point of this section:
>  1. As shown in Appendix B, embedding-level spectral analysis has not been investigate yet, and directly analyze spectrum on embeddings is meaningless. The observation in Section 3, though seems simple, **is the key point that make the analysis possible**. Just because the sequence-level representation contains the sample-level temporal structure, we can apply spectral analysis on the embeddings whose temporal structure is disrupted on sample-level. We believe this observation will further assist in generating new analyses and discoveries.
>  2. We absolutely agree with you that adjacent sequence spectral analysis is a general framework for time series forecasting models, which can be used to analyze the spectral features of embeddings in most SOTA deep learning models like DLinear or XXXformer as long as they sample time series in sliding window manner. Also, we further apply this framework to a model with an MLP-based backbone.
>  3. The condition of "a linear transformation" in our theoretical analysis **aligns with the structure of most current models**, where the embedding is mapped to the final output merely through a linear layer called the prediction head. Also, a challenge is to **align the gap between sample-level and sequence-level frequency domain representations**. To address this, we use stochastic processes as the framework for analysis. The results of this analysis indicate a direct correlation between the frequency collapse in the embedding sequence and the underfitting in the prediction results.
>  4. Due to page limitations we present one case in Section 4. More cases can be found in the Appendix section: "MORE COMPARISON OF THE THREE MODELS". It's worth noting that in Fig.6 the low-pass filter is applied on sequence-level representations, so the results show how the change of sequence-level representations affects the sample-level representations.
>  5. Please refer to the section \#"Weakness in Empirical Study".
>
> ## \# Weakness in *Presentation*
>
> Thanks for your valuable suggestions. Your suggestions greatly enhance the completeness of our work. We review the recent works on spectral analysis in Appendix **"Supplementary Information on Spectral Analysis"**, and some other works regarding the theory of frequency collapse are listed in Appendix C: **"Supplementary Analysis on Frequency Collapse"**. For details please refer to the revised paper.
>
> ## \# Weakness in *Empirical evaluation*
> Your suggestions are highly valuable and align with broader concerns. In the newly added section in the revised paper, we address your points from two aspects:
> 1. **Introducing a Quantitative Metric (Appendix C.2)**. We propose a new quantitative metric to measure the richness of mid-to-high-frequency information in the model embeddings.
> 2. **Evaluating the Effectiveness of the PErM Strategy on More Models (Appendix D.2)**. We assess the impact of the PErM strategy on more kinds of baselines.
>
> Experimental results demonstrate that when training time series foundation models, both transformer-based and MLP-based models exhibit varying degrees of frequency collapse. Additionally, our supplementary experimental results confirm that the PErM strategy also improves the performance of MLP-based models. Here are the summarized results, and for a detailed description please refer to our revised paper.
> |   MLP-backbone Variants   | w/o PErM | with PErM |
> | ---- | ---- | ---- |
> | Avg. MSE |   0.264   |  0.253  |
>
> And thanks for your suggestion for reporting standard deviations. Since we select 17 baselines and 28 settings for each baseline, it may take some more time to obtain the results.

---

> ### Author Response · Authors · 2024-11-21
> **Response to your concerns (Part 2/2)**
>
> ## \# Weakness in "Minor comments"
>
> **1-1**.
> We remove the positional embedding (PE) since the backbone without PE gets better prediction results. Here are the experimental results:
>
> |   Variants   | with PE | without PE   |
> | ---- | ---- | ---- |
> | Avg. MSE |   0.273   |   0.268   |
>
> Moreover, under the assumption of AutoFormer[1], autocorrelation plays an important role in time series analysis. Directly adding PE to the time series is functionally equivalent to adding a trend component to the time series, which strengthens the non-stationary feature of the series, making it more difficult for the model to learn the autocorrelated properties within the time series.
>
> **2-1**.
> This statement pertains to models where node interchangeability occurs during the forward process (e.g., linear mapping, attention operations). Node interchangeability implies that the positions of elements in the embedding can be arbitrarily rearranged, resulting in the disruption of the original temporal structure. We present the detailed information in Appendix B.2 **Challenges in Embedding-level Spectral Analysis**.
>
> **2-2**.
> Thanks for your suggestion, as mentioned in response to "Weakness in *Significance and soundness*-1", this is a simple but insightful observation. So how about using "observation"?
>
> **2-3 & 2-4 & 2-5**.
> We kindly recognize these points as a supplement to the previous issues, and our response to previous questions includes the response to these points. Also, if you have any unsolved questions, we are delighted to have a further discussion with you.
>
> ***If you have any other questions or have some insights to share, we welcome further discussion with you.***
>
> [1] Wu, Haixu, et al. "Autoformer: Decomposition transformers with auto-correlation for long-term series forecasting." Advances in neural information processing systems 34 (2021): 22419-22430.

---

> > ### Comment · Reviewer_nD11 · 2024-11-25
> >
> > Thanks for the rebuttal, but my main concerns remain.
> >
> > I still think the paper has severe issues regarding analysis, presentation, and overstated claims as detailed in the review.
> >
> > Adding a few appendices is not enough to fix these issues. While I believe that some ideas are potentially interesting, I think the paper needs a complete re-work.
> >
> > I will keep my score.

---

### Official Review · Reviewer_Us8Q · 2024-11-06

**Soundness:** 3
**Presentation:** 2
**Contribution:** 2
**Rating:** 5
**Confidence:** 2

**Summary:**

This paper addresses the frequency collapse issue observed in Transformer-based time series forecasting models, where the embeddings generated by the model lack high-frequency signals, leading to underfitting and poor prediction performance. To tackle this problem, the authors propose the Post-Embedding ReMapping (PErM) strategy. PErM introduces a predefined non-linear remapping layer between the top layer of the Transformer backbone and the model's prediction head. This layer projects high-frequency features from the original signals into a new frequency domain while re-integrating information within the embeddings. Experimental results demonstrate that using various PErM functions effectively alleviates the frequency collapse phenomenon and significantly improves the final prediction performance.

**Strengths:**

1. The paper identifies and addresses a novel issue, frequency collapse, in Transformer-based time series forecasting models. The proposed PErM strategy is an innovative solution that uses a non-linear remapping layer to enhance the frequency-domain representation of embeddings, offering a fresh perspective on improving model performance.
2. The authors provide thorough experimental validation of the PErM strategy. They conduct experiments on multiple datasets and compare their method with several state-of-the-art approaches, demonstrating the effectiveness of PErM in enhancing prediction accuracy. The experimental setup is robust and the results are compelling.
3. The paper is well-structured and clearly written.

**Weaknesses:**

1. The authors claim that transformers can lead to a lack of high-frequency signals in the embeddings, but they do not provide a theoretical explanation for this phenomenon. A deeper theoretical analysis would help understand why transformers tend to produce embeddings with fewer high-frequency components.
2. The authors propose that using various PErM functions can effectively alleviate the frequency collapse phenomenon and support this claim with experimental results. However, they do not provide a clear and reasonable explanation for why PErM functions are effective in mitigating frequency collapse. A more detailed theoretical or empirical analysis would strengthen the understanding of the underlying mechanisms.
3. Some figures in the paper are not referenced or explained in the main text. This makes it difficult for readers to understand the purpose and meaning of these figures. Adding references to these figures and providing detailed explanations would enhance the readability and clarity of the paper.
4. The experimental section seems to lack some commonly used datasets, such as Electricity and Exchange. Including these datasets would provide a more comprehensive evaluation of the proposed method and allow for better comparison with existing state-of-the-art models.
5. From the results, it appears that PErMformer performs better in shorter horizons and on the ETTm2 dataset for longer horizons. The paper does not provide a theoretical explanation for this variability. Using the theoretical framework presented in the paper, a detailed analysis of why PErMformer excels in different horizon lengths could provide valuable insights and improve the understanding of the method's strengths and limitations.

**Questions:**

Refer to the weakness.

---

> ### Author Response · Authors · 2024-11-21
> **Response to your concerns**
>
> Thanks for your comprehensive and valuable review. We will address your concerns one by one.
>
> ## \# Weakness 1:
> We appreciate the reviewer’s valuable suggestions, and we added the section **"Supplementary analysis on frequency collapse"** in the Appendix (Appendix C).
>
> Specifically, Luo et.al.[1] theoretically proved that compared to low-frequency signals, high-frequency signals are more difficult to learn, and the gradient of high-frequency signals has a theoretical upper bound during the gradient descent process in multi-layer networks with L2 loss function. We believe this theory can explain the frequency collapse phenomenon to some extent despite that the black-box transformer backbone is currently too complex to be analyzed theoretically.
>
> ## \# Weakness 2:
> Thanks for the valuable insights. We added section **"Supplementary analysis on PErM strategy"** in Appendix D to address your concerns.
>
> In brief, **the PErM strategy helps the model to learn the high-frequency information from a low-frequency perspective**. Specifically, PErM functions can introduce more high-frequency components and potentially result in harmonic components. Conversely, this implies that high-frequency signals in the final embedding can be represented by lower-frequency signals before the PErM layer. According to F-principle, these lower-frequencies are easier to learn. In this way, the model can eventually model the signals more precisely. Our further quantitive experiments empirically prove that PErM is effective not only in transformer-based models, but also in other models. For detailed information please refer to the added section **"Supplementary analysis on PErM strategy"** in Appendix C.
>
> ## \# Weakness 3:
> Quite good advice for improving our writing. We have fixed these problems in the revised version, and each figure is explained in detail within the main text.
>
> ## \# Weakness 4:
> The ECL dataset in our benchmark is an alias for the dataset Electricity you mentioned. We are running all baselines on dataset Exchange, and obtaining the results for all 17 baselines may take some more time.
>
> ## \# Weakness 5:
> This is an interesting finding. We can also find that some methods like FITS and UniTime exhibit similar phenomena on datasets such as ETTh1. These cases may be caused by a combination of model training uncertainties and the characteristics of the data.
>
> ***If you have any other questions or have some insights to share, we welcome further discussion with you.***
>
> [1] Luo, Tao, et al. "Theory of the frequency principle for general deep neural networks." arXiv preprint arXiv:1906.09235 (2019).

---

> > ### Comment · Reviewer_Us8Q · 2024-11-26
> >
> > I have reviewed the authors' rebuttal, and while they have addressed most of my concerns, I will maintain my current score due to the paper's quality, novelty, and presentation.

---

### Author Response · Authors · 2024-11-21
**Revised paper is updated**

We have submitted the revised paper. 3 Supplement Sections have been added to the Appendix (Appendix B,C,D), including supplementary information on spectral analysis,  supplementary analysis on frequency collapse, and supplementary analysis on PErM strategy. The text of updated section has been set to blue. We hope these sections can address some of your concerns.

---

### Author Response · Authors · 2024-11-25
**Looking forward to your reply**

Dear reviewers,

Thank you for your time and constructive feedback. We have provided specific responses to your comments after carefully considering your suggestions and concerns. We hope our explanations will dispel your concerns.

We would be happy to receive more valuable comments and have further discussions with you. We look forward to hearing from you soon!

---

### Note · Authors · 2024-11-28

I have read and agree with the venue's withdrawal policy on behalf of myself and my co-authors.